# Partitioning Message Passing for Graph Fraud Detection

**Wei Zhuo[1], Zemin Liu[2], Bryan Hooi[2], Bingsheng He[2]\*, Guang Tan[1]\*, Rizal Fathony[3], Jia Chen[3]**

[1]Shenzhen Campus of Sun Yat-sen University, [2]National University of Singapore,
[3]GrabTaxi Holdings Pte. Ltd.

`zhuow5@mail2.sysu.edu.cn, zeminliu@nus.edu.sg,`
`{bhooi, hebs}@comp.nus.edu.sg, tanguang@mail.sysu.edu.cn,`
`{rizal.fathony, jia.chen}@grab.com`

## Abstract

Label imbalance and homophily-heterophily mixture are the fundamental problems encountered when applying Graph Neural Networks (GNNs) to Graph Fraud Detection (GFD) tasks. Existing GNN-based GFD models are designed to augment graph structure to accommodate the inductive bias of GNNs towards homophily, by excluding heterophilic neighbors during message passing. In our work, we argue that the key to applying GNNs for GFD is not to exclude but to *distinguish* neighbors with different labels. Grounded in this perspective, we introduce Partitioning Message Passing (PMP), an intuitive yet effective message passing paradigm expressly crafted for GFD. Specifically, in the neighbor aggregation stage of PMP, neighbors with different classes are aggregated with distinct node-specific aggregation functions. By this means, the center node can adaptively adjust the information aggregated from its heterophilic and homophilic neighbors, thus avoiding the model gradient being dominated by benign nodes which occupy the majority of the population. We theoretically establish a connection between the spatial formulation of PMP and spectral analysis to characterize that PMP operates an adaptive node-specific spectral graph filter, which demonstrates the capability of PMP to handle heterophily-homophily mixed graphs. Extensive experimental results show that PMP can significantly boost the performance on GFD tasks. Our code is available at `https://github.com/Xtra-Computing/PMP`.

## 1 Introduction

With the explosive growth of online information, fraudulent activities have significantly increased in financial networks (Ngai et al., 2011; Lin et al., 2021), social media (Deng et al., 2022), review networks (Rayana & Akoglu, 2015), and academic networks (Cho et al., 2021), making the detection of such activities an area of paramount importance. To fully exploit the rich graph structures contained in fraud graphs, recent studies have increasingly adopted Graph Neural Networks (GNNs) (Wu et al., 2020) to Graph Fraud Detection (GFD).

Applying message passing GNNs (Gilmer et al., 2017) in GFD encounters two significant challenges: label imbalance (Liu et al., 2023) and a mixture of heterophily and homophily (Gao et al., 2023a). Network attackers often employ sophisticated tactics to mimic regular network patterns, by strategically injecting a limited number of fraud nodes in the main contexts of the target graph to hide their fraudulent activities. The label imbalance problem causes the GNN to primarily capture the patterns and characteristics of benign nodes, compromising their ability to accurately identify fraudulent ones. Additionally, the heterophily-homophily mixture violates the homophily (Zhu et al., 2020) inductive bias of GNNs, as fraud nodes are often strategically placed within benign communities to exhibit heterophily, while the context around benign nodes exhibits homophily. Therefore, it is pivotal to develop GNNs that can navigate these challenges adeptly.

To alleviate these issues, several methods have been proposed from a spatial perspective to diminish the impact of heterophilic neighbors during the aggregation process. These strategies commonly involve utilizing a trainable approach or predetermining a mechanism to resample neighbors (Dou et al., 2020; Liu et al., 2021c; 2020) or reweight edges (Wang et al., 2019; Cui et al., 2020; Shi et al.,

---

\*Corresponding authors.

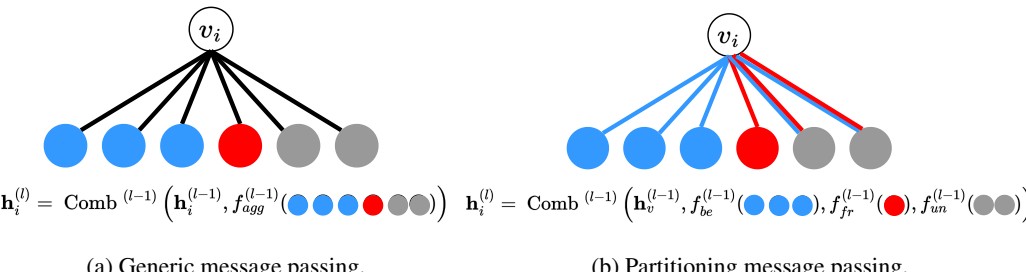

(a) Generic message passing.

(b) Partitioning message passing.

Figure 1: Comparison between generic message passing GNN and partitioning message passing GNN. Red, blue and grey mean fraud, benign and unlabeled nodes, respectively.

2022; Liu et al., 2021a). From a spectral view, GHRN (Gao et al., 2023a) shows using a high-pass filter helps identify heterophilic neighbors, pruning inter-class edges to highlight the high frequencies of the graph. BWGNN (Tang et al., 2022) reveals the 'right-shift' phenomenon of spectral energy and suggests a Beta wavelet-based band-pass graph filter.

However, from the spatial perspective, augmenting the graph structure to exclude heterophilic neighbors during message passing to accommodate the inductive bias of GNNs is non-trivial for two reasons. Firstly, semi-supervised GFD usually makes pruning or reweighting of unlabeled neighbors based on predicted logits unreliable (Gao et al., 2023c). In other words, the edge weight or pruning probability depends on the similarity of the predicted representations between nodes, therefore any prediction error may accumulate and impact the final result. Secondly, these methods face scalability issues on large-scale graphs. On the other hand, spectral-based models can keep the graph fixed and learn the spectral graph filter beyond low-pass (Bo et al., 2021; He et al., 2021; Tang et al., 2022; Gao et al., 2023c), which demonstrate efficacy in addressing heterophily. However, these models usually struggle with label imbalance due to shared parameters and require the entire graph input, hindering the mini-batch training.

In this paper, we argue that complicated trainable or predetermined strategies for excluding heterophilic neighbors are unnecessary. Instead, the key of applying GNNs on fraud graphs is to *distinguish* neighbors during the message passing process, rather than exclusion. A powerful model should inherently have the capacity to adaptively modulate the information derived from both homophilic and heterophilic neighbors.

Based on this insight, we propose an intuitive yet effective message passing paradigm named Partitioning Message Passing (PMP). At the neighbor aggregation stage of PMP, we employ distinct aggregation functions to independently handle homophilic and heterophilic neighbors. It ensures that parameter sharing is confined to nodes within the same class, thereby preventing the gradients from being dominated by the majority class nodes. For neighbors with unknown classes, we configure their aggregation function to be a flexible composite of the aggregation functions used for labeled neighbors, where the combination weight is derived from an adaptable scalar function unique to the center node. Besides, since the fraud graph is homophily-heterophily mixed, the amount of information that different nodes obtain from homophilic neighbors and heterophilic neighbors should also be adaptively adjusted according to the node. To achieve it, we treat the parameter matrices of aggregation functions as the output of weight generators with respect to the center node. By this means, each node can adaptively adjust the influence from different classes of neighbors without additional parameters. Moreover, our theoretical analysis for PMP bridges the gap between the spatial form of the model and the spectral explanation, which proves that PMP can be interpreted as a node-specific spectral convolution, i.e., each node has its own spectral graph filter. It shows that our model is more suitable for graphs with homophilic and heterophilic mixtures because the uniform graph filter can not balance both aspects. Extensive experiments on four benchmark datasets demonstrate the effectiveness of PMP.

## 2 BACKGROUND AND MOTIVATION

### 2.1 BACKGROUND

**Notations.** A multi-relational attributed fraud graph is denoted as $\mathcal{G} = (\mathcal{V}, \{E_r\}_{r=1}^{R}, \mathbf{X}, \mathcal{Y})$, where $\mathcal{V} = \{v_1, \cdots, v_N\}$ represents the node set with $N$ nodes, $E_r$ represents the edge set under the

$r$-relation, and $R$ is the total number of relations. Thus, $G$ can be seen as a combination of $r$ separate single-relational graphs, each of which is characterized by an adjacency matrix $\mathbf{A}_r \in \mathbb{R}^{N \times N}$ derived from its corresponding edge set $E_r$, i.e., if an edge $e_{ij} \in E_r$, $\mathbf{A}_{r,ij} = 1$; otherwise $\mathbf{A}_{r,ij} = 0$. $\mathbf{X} \in \mathbb{R}^{N \times d}$ is a node feature matrix, wherein the $i$-th row $\boldsymbol{x}_i$ is the feature vector of $v_i$. $\mathcal{Y}$ represents the set of labels, where each node $v_i$ is assigned a binary label $y_i \in \mathcal{Y}$, i.e., $y_i = 1$ denotes fraud node, and 0 denotes benign node. Due to the label imbalance nature in fraud graphs, the quantity of benign nodes (the majority class) substantially outweighs that of the fraud ones (the minority class). For a node $v_i$ and its neighbor $v_j$, if $y_i = y_j$, $v_j$ is a homophilic neighbor; otherwise, it is heterophilic.

**Graph Neural Networks.** In a typical GNN, the $l$-th message passing layer for a node $v_i$ operates by iteratively aggregating data from its immediate neighbors, and then combining the aggregated messages with its own node representation (Gilmer et al., 2017):

$$\mathbf{h}_i^{(l)} = \mathrm{Comb}^{(l-1)} \left( f_{\mathrm{self}}^{(l-1)} \left( \mathbf{h}_i^{(l-1)} \right), f_{\mathrm{agg}}^{(l-1)} \left( \left\{ \mathbf{h}_j^{(l-1)} | v_j \in \mathcal{N}(v_i) \right\} \right) \right), \tag{1}$$

where $\mathbf{h}_i^{(l)}$ is the $l$-th layer hidden representation of $v_i$, $\mathrm{Comb}(\cdot)$ is a combination function, $f_{\mathrm{self}}(\cdot)$ is a function on the center node, $f_{\mathrm{agg}}(\cdot)$ is permutation invariant neighbor aggregation function, and $\mathcal{N}(v_i)$ is the neighbor set of $v_i$. $f_{\mathrm{agg}}(\cdot)$ typically encompasses two essential components: feature transformation and message fusion. Existing GNNs have primarily focused on elaborating on message fusion functions based on Laplacian (Kipf & Welling, 2017; Defferrard et al., 2016), diffusion (Gasteiger et al., 2019a;b; Chien et al., 2021), attention (Veličković et al., 2018), or a combination of multiple aggregators (Corso et al., 2020). However, these methods adopt simple linear feature transformations by employing weight matrices shared across all neighbors. This can lead to a learning process that is biased by the potential label imbalance. The backpropagation of gradients, which aims to minimize the overall loss, tends to be dominated by the majority class neighbors. This limits the ability of the model to learn from the minority class effectively. If the center node belongs to a minority class, the graph embodies heterophily. Such interplay of *heterophily* and *label imbalance* in fraud graphs provides a unique challenge for GNNs.

**Related Work.** (1) **Label Imbalanced Learning on Graphs.** GNNs are known to be sensitive to label imbalance (Liu et al., 2023). Currently, several studies have proposed different strategies to address this challenge, such as by implementing adversarial constraints (Shi et al., 2020), generating minority instances using techniques like GAN (Qu et al., 2021) or SMOTE (Zhao et al., 2021), and modifying the degree of imbalance compensation (Song et al., 2022). Despite their effectiveness, their success heavily relies on augmenting graph structure to balance the label distribution, which can hardly be adaptive to fraud detection tasks on large-scale graphs. (2) **Graph Heterophily Learning.** Heterophily has emerged as a significant concern for GNNs. This issue was initially highlighted by Pei et al. (2020). Separating ego- and neighbor-embeddings (Zhu et al., 2020; Platonov et al., 2023; Hamilton et al., 2017) proves to be an effective technique for learning on heterophilic graphs. Given that nodes sharing the same class are distantly placed within heterophilic graphs, several approaches aim to extend the local neighbors to non-local ones by integrating multiple layers (Chien et al., 2021; Abu-El-Haija et al., 2019), and identifying potential neighbors through attention mechanisms (Liu et al., 2021b; Yang et al., 2022b) or similarity measures (Zhuo & Tan, 2022; Jin et al., 2021). Spectral-based methods (Luan et al., 2021) overcome this challenge by introducing additional graph filters and mixture strategy, which aims to adaptively integrate information by emphasizing certain frequencies. This approach shares a similar goal with our work. We discuss in detail how our model relates to this method in Appendix C. (3) **GNN-based Fraud Detection.** GNNs have been leveraged to detect fraudulent activities in financial services (Rao et al., 2022; Xu et al., 2021; Lin et al., 2021; Chen et al., 2024), telecommunications (Nabeel et al., 2021), social networks (Deng et al., 2022), and healthcare (Cui et al., 2020). CARE-GNN (Dou et al., 2020) employs a label-aware similarity measure to identify informative neighbors and leverages reinforcement learning to selectively integrate similar neighbors. PC-GNN (Liu et al., 2021c) utilizes label-balanced samplers for sub-graph training. Spectral-based methods (Tang et al., 2022; Gao et al., 2023c) conduct spectral analysis on fraud graphs, designing band-pass or high-pass graph filters specifically tailored to detect fraud nodes. GAGA (Wang et al., 2023) groups neighbors by augmenting neighbor features with their labels explicitly and treating unlabeled neighbors as a new class.

## 2.2 MOTIVATION ANALYSIS

To probe the reasons why message passing based GNNs experience shortcomings in fraud detection tasks, our investigation is conducted through an analysis of the mutual influence between nodes resulting from the message passing process. Inspired by (Xu et al., 2018; Zhang et al., 2021), the influence of node $v_j$ on the center node $v_i$ can be quantified by measuring how alterations in the input feature of $v_j$ affect the representation of $v_i$ after $k$ iterations of message passing. For any $v_i$ and its neighbor $v_j \in \mathcal{N}(v_i)$, given the message passing form $\mathbf{H}^{(k)} = \hat{\mathbf{A}}^k \mathbf{H}^{(0)} \mathbf{W}$ where $\mathbf{H}^{(0)} = \mathbf{X}$, considering the $h$-th feature of $\mathbf{X}$, the influence of $v_j$ on the final representation of $v_i$ is defined as:

$$I(k)_{ij} = \left[ \frac{\partial \mathbf{H}_{ih}^{(k)}}{\partial \mathbf{H}_{jh}^{(0)}} \right]_{\forall h \in \{1, \cdots d\}} = \hat{\mathbf{A}}_{ij}^k \mathbf{W}. \tag{2}$$

Since the gradient is independent of the feature dimension $h$ (Zhang et al., 2021), the final result omits the $h$. In a fraud graph characterized by imbalanced label distribution, let $m$ represent the number of benign neighbors $\mathcal{N}_{\text{be}}(v_i)$ of node $v_i$, and $n$ represents the number of fraud neighbors $\mathcal{N}_{\text{fr}}(v_i)$. In this context, $m \gg n$. Since we specifically study the class imbalance problem, to rule out potential interference from other variables, we assume the graph to be regular. Consequently, all non-zero off-diagonal entries in $\hat{\mathbf{A}}^k$ are assumed to be equal, and this common value is denoted by $\gamma$. The total influence of benign neighbors on the center node $v_i$ is $I_{\mathcal{N}_{\text{be}}(v_i) \to v_i} = \frac{1}{\gamma} \sum_{j \in \mathcal{N}_{\text{be}}(v_i)} \hat{\mathbf{A}}_{ij}^k w = m\mathbf{W}$, where we use $\gamma$ to scale the influence score. Similarly, for fraud neighbors we have $I_{\mathcal{N}_{\text{fr}}(v_i) \to v_i} = n\mathbf{W}$. Then, the total influence from neighbors is given by $(m + n)\mathbf{W} \approx I_{\mathcal{N}_{\text{be}}(v_i) \to v_i}$, which tends to over-amplify the influence from the majority class neighbors (benign) while neglecting that from the minority class neighbors (fraudulent). Such an imbalance in influence can skew the message passing process, causing it to be insufficiently responsive to the nuances of the minority class nodes. As a result, the capacity of the network to capture critical features from the minority class neighbors diminishes, potentially undermining its effectiveness in fraud detection especially when the graph exhibits heterophily.

To address the aforementioned problems, many existing GNN-based fraud detection models adopt strategies such as re-weighting neighbors from different classes using predefined indicators (Liu et al., 2021c) or learnable attention values (Wang et al., 2019; Liu et al., 2021a; Shi et al., 2022; Cui et al., 2020). Additionally, they often incorporate a learnable sampler to selectively focus on potential neighbors belonging to the same class as the center node (Dou et al., 2020; Liu et al., 2020). These works typically utilize additional modules to augment the graph structure, aiming to homogenize contextual information in graphs. The essential goal of these methods is to modify the graph to exclude heterophilic neighbors (i.e., neighbors that belong to different classes than the center node), thereby tailoring the graph topology for more effective processing by GNNs in fraud detection tasks. However, augmenting the graph structure often leads to high time and memory complexity, limiting scalability to large graphs. In our work, we argue that adapting the graph to fit GNNs is both costly and unnecessary, but keeping the graph fixed and modifying the GNN model to fit the graph can be more efficient and effective.

Upon observing the formation of the total influence $(m + n)\mathbf{W}$, we find that both fraud and benign neighbors are weighted equally with $\mathbf{W}$, which leads to benign neighbors dominating the gradient of $\mathbf{W}$ during the backpropagation. It naturally motivates us to ask whether handling neighbors of two distinct classes with separate weight matrices might allow for adaptive adjustment in their influence on the center node, i.e., $m\mathbf{W}_1 + n\mathbf{W}_2$. Such an approach may effectively mitigate issues related to label imbalance and heterophily, without doing any operations on the graph itself.

## 3 PARTITIONING MESSAGE PASSING

Our preliminary analysis in Section 2.2 reveals a relationship between parameter sharing within GNNs and biases in the learning process. In this section, we formally present our method, Partitioning Message Passing (PMP), which is a simple, intuitive, yet powerful approach tailored for the fraud detection task. The basic idea of PMP is to utilize the label information to partition the message passing process, enabling the model to distinguish neighbors according to their classes by learning different weights for each class during message passing, thereby enhancing its ability to adaptively adjust the influence propagated from class-imbalanced neighboring nodes. As shown in Fig. 1b, the

$l$-th message passing iteration of PMP for a node $v_i$ is described as follows:

$$\mathbf{h}_i^{(l)} = \text{Comb}^{(l-1)} \Bigg( f_{\text{self}}^{(l-1)} \left( \mathbf{h}_i^{(l-1)} \right), f_{\text{fr}}^{(l-1)} \left( \mathbf{h}_j^{(l-1)} \mid v_j \in \mathcal{N}_{\text{fr}}(v_i) \right),$$

$$f_{\text{be}}^{(l-1)} \left( \mathbf{h}_j^{(l-1)} \mid v_j \in \mathcal{N}_{\text{be}}(v_i) \right), f_{\text{un}}^{(l-1)} \left( \mathbf{h}_j^{(l-1)} \mid v_j \notin \mathcal{N}_{\text{be}}(v_i) \cup \mathcal{N}_{\text{fr}}(v_i) \right) \Bigg), \tag{3}$$

where $f_{\text{fr}}^{(l)}(\cdot)$, $f_{\text{be}}^{(l)}(\cdot)$, and $f_{\text{un}}^{(l)}(\cdot)$ correspond to the handling of neighbors categorized as fraud, benign, and unlabeled, respectively. These functions are parameterized by the weight matrices $\mathbf{W}_{\text{fr}}^{(l)}$, $\mathbf{W}_{\text{be}}^{(l)}$ and $\mathbf{W}_{\text{un}}^{(l)} \in \mathbb{R}^{d \times d'}$, aligning with the aforementioned categories. We simply set the message fusion of aggregation functions to *sum*: $f_{\text{fr}}^{(l-1)}(\mathbf{h}_j^{(l-1)} | v_j \in \mathcal{N}_{\text{fr}}(v_i)) = \sum_{v_j \in \mathcal{N}_{\text{fr}}(v_i)} \mathbf{h}_j^{(l-1)} \mathbf{W}_{\text{fr}}^{(l-1)}$.

**Handling unlabeled neighbors as a weighted combination of fraud and benign labels.** For unlabeled neighbors, we face the challenge of determining an appropriate transformation that acknowledges the uncertainty of these connections. Treating $f_{\text{un}}$ for unlabeled neighbors independently from those for fraud and benign neighbors is not suitable, as it would mean forcing an additional, definite label on these neighbors. For binary classification fraud detection task, unlabeled neighbors share characteristics with both benign and fraud categories, while a separate, definite label might make it harder to model the continuous spectrum between clear-cut benign and fraud behavior and can not leverage the knowledge captured by $f_{\text{fr}}$ and $f_{\text{be}}$. Instead, PMP aims to treat unlabeled neighbors in a way that reflects their uncertain and mixed nature. Thus we define $\mathbf{W}_{\text{un}}^{(l)}$ as a weighted combination of $\mathbf{W}_{\text{fr}}^{(l)}$ and $\mathbf{W}_{\text{be}}^{(l)}$:

$$\mathbf{W}_{\text{un}}^{(l)} = \alpha_i^{(l)} \mathbf{W}_{\text{fr}}^{(l)} + (1 - \alpha_i^{(l)}) \mathbf{W}_{\text{be}}^{(l)}, \tag{4}$$

where $\alpha_i^{(l)} \in \mathbb{R}^{(0,1)}$ is a scalar to modulate the class tendency of unlabeled neighbors associated with the center node $v_i$. In other words, a large $\alpha_i^{(l)}$ means that the model treats unlabeled neighbors more similarly to fraud nodes, indicating a tendency or suspicion toward fraud for those particular connections, and vice versa. The parameter $\alpha_i^{(l)}$ thus serves as a continuous dial that allows the model to smoothly interpolate between treating unlabeled neighbors as either benign or fraud nodes, by dynamically adjusting $\alpha_i^{(l)}$ based on characteristics of each center node $v_i$. Since $\alpha_i^{(l)}$ should be node-specific and adaptive, we define $\alpha_i^{(l)}$ as a function of $v_i$, $\alpha_i^{(l)} = \Phi(\mathbf{h}_i^{(l-1)})$, where $\Phi : \mathbb{R}^d \to \mathbb{R}$ is a single-layered MLP with a sigmoid activation shared across all nodes.

**Root-specific weight matrix generation.** In Eq. (3), the influence from the majority and minority class neighbors on $v_i$ can be adaptively adjusted by $\mathbf{W}_{\text{be}}$ and $\mathbf{W}_{\text{fr}}$ respectively. Following the analysis in Section 2.2, the total influence from labeled neighbors on $v_i$ is $m\mathbf{W}_{\text{be}} + n\mathbf{W}_{\text{fr}}$, where $\mathbf{W}_{\text{be}}$ and $\mathbf{W}_{\text{fr}}$ are shared across all center nodes when performing message passing. Further, in various contexts across the graph, nodes often exhibit unique characteristics and patterns of interactions. These differences can arise from distinct node attributes or their positional contexts within the graph structure. Given this variability, a uniform approach to regulating the influence from different classes of neighbors might oversimplify these interactions. Instead, a more adaptive and node-specific method would better capture the intricate dynamics between a node and its diverse neighborhoods. However, directly equipping every center node with individualized weight matrices, $\mathbf{W}_{\text{fr}}^{(l)}$ and $\mathbf{W}_{\text{be}}^{(l)}$, is intractable. To circumvent this, we introduce learnable weight generators that model $\mathbf{W}_{\text{fr}}^{(l)}$ and $\mathbf{W}_{\text{be}}^{(l)}$ as functions of the center node. Importantly, while these generators produce node-specific weight matrices, their underlying parameters are shared across all center nodes. Such design strikes a balance between adaptability and model compactness, eliminating the risk of an exponential surge in parameter count. Specifically, for a center node $v_i$, the $(l+1)$-th layer weight matrices of its aggregation functions are defined as:

$$\mathbf{W}_{\text{fr},i}^{(l)} = \Psi_{\text{fr}}(\mathbf{h}_i^{(l)}) \qquad \mathbf{W}_{\text{be},i}^{(l)} = \Psi_{\text{be}}(\mathbf{h}_i^{(l)}), \tag{5}$$

where $\Psi_{\text{fr}}, \Psi_{\text{be}} : \mathbb{R}^d \to \mathbb{R}^{d \times d'}$ are two learnable weight generators defined as $\Psi_{\text{fr}}(x) = \text{MLP}_{\text{fr}}(\text{diag}(x))$ and $\Psi_{\text{be}}(x) = \text{MLP}_{\text{be}}(\text{diag}(x))$, and they are both implemented as single-layered MLPs; $\text{diag}(x)$ is the diagonal matrix of the input vector. Each node $v_i$ can receive a distinct $\mathbf{W}_{\text{fr},i}^{(l)}$

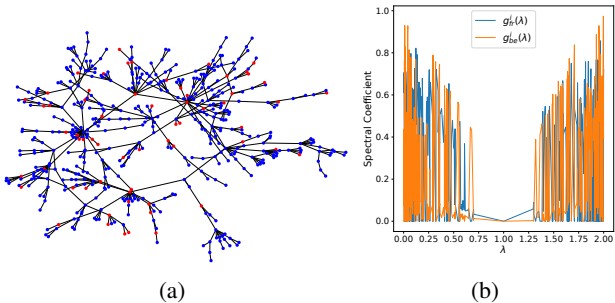

(a)                                 (b)

Figure 2: (a) A Barabási–Albert graph with 500 nodes, with 10% nodes are fraud. Features of benign nodes (depicted in blue) follow a Gaussian distribution, $\mathcal{N}(1,1)$, while those of fraud nodes (shown in red) are drawn from $\mathcal{N}(5,1)$. (b) Spectral convolution filters of PMP for a random node $v_i$.

and $\mathbf{W}_{be,i}^{(l)}$ tailored to its unique features, while the generation process $\mathrm{MLP_{fr}}$ and $\mathrm{MLP_{be}}$ remains consistent and is governed by shared parameters across all nodes.

To summarize, PMP works as Algorithm 1 of Appendix A under the $r$-th relation, utilizing mini-batch training. For $R$-relational graphs, we perform PMP propagation for each relation separately, resulting in $R$ representations for each node. We then add a concatenation pooling followed by an MLP to integrate $R$ node representations for each node to produce the final node representation.

## 4 THEORETICAL INSIGHTS

As mentioned in Section 2.1, graphs with fraud usually exhibit a heterophily-homophily mixture, as attackers would like to sparsely inject a limited number of fraud nodes into benign communities to camouflage their activities and spread influence. For regions exhibiting homophily, where the center node and its neighbors predominantly share the same label, the desired GNN should act as a loss-pass filter to smooth the feature representations in the locality. Conversely, in heterophilic regions, where the center node's label diverges from most of its neighbors, the GNN must adaptively shift its spectral response to capture such contrasting label information. Some work (Yang et al., 2022a; Wang & Zhang, 2022) show that assigning each feature dimension a separate spectral filter improves the performance of GNNs. Differently, our model achieves adaptivity at the node level, with each node being assigned a separate spectral filter. Specifically, we present the following theorem.

**Theorem 1.** *Consider an undirected graph $\mathcal{G}$, let $\mathbf{L} = \mathbf{U}\mathbf{\Lambda}\mathbf{U}^\top$ represent the eigendecomposition of the symmetric normalized Laplacian $\mathbf{L} = \mathbf{I} - \mathbf{D}^{-1/2}\mathbf{A}\mathbf{D}^{-1/2}$, where $\mathbf{U}$ is the matrix of eigenvectors and $\mathbf{\Lambda} = \mathrm{diag}([\lambda_i]_{i=1\ldots N})$ is the diagonal matrix of eigenvalues and $0 = \lambda_1 \leq \cdots \leq \lambda_n \leq 2$. Given two sets of $d'$ graph signals $\mathbf{X}\mathbf{W}_{fr}$ and $\mathbf{X}\mathbf{W}_{be}$, PMP scheme described in Eq. (3) operates as an adaptive graph filter, with the node-specific spectral convolution for node $v_i$ on $\mathbf{X}\mathbf{W}_{fr}$ and $\mathbf{X}\mathbf{W}_{be}$ is given as:*

$$\mathbf{H}(v_i) = g_{fr}^i(\mathbf{L})\mathbf{X}\mathbf{W}_{fr} + g_{be}^i(\mathbf{L})\mathbf{X}\mathbf{W}_{be} = \mathbf{U}g_{fr}^i(\Lambda)\mathbf{U}^\top\mathbf{X}\mathbf{W}_{fr} + \mathbf{U}g_{be}^i(\Lambda)\mathbf{U}^\top\mathbf{X}\mathbf{W}_{be} \quad (6)$$

*where the spectral convolution filters are diagonal matrices defined as:*

$$g_{fr}^i(\Lambda)[j,j] = \begin{cases} 1-\lambda_j & v_j \in \mathcal{N}_{fr}(v_i) \\ 0 & v_j \in \mathcal{N}_{be}(v_i) \\ \alpha_i(1-\lambda_j) & \text{otherwise} \end{cases} \qquad g_{be}^i(\Lambda)[j,j] = \begin{cases} 0 & v_j \in \mathcal{N}_{fr}(v_i) \\ 1-\lambda_j & v_j \in \mathcal{N}_{be}(v_i) \\ (1-\alpha_i)(1-\lambda_j) & \text{otherwise} \end{cases}.$$
$$(7)$$

*where $\mathcal{N}_{fr}(v_i)$ and $\mathcal{N}_{be}(v_i)$ are respectively the fraud neighbors and benign neighbors of $v_i$ in the training set. The $i$-th row of the matrix $\mathbf{H}(v_i)$, i.e., $\mathbf{H}(v_i)[i,:]$, is the representation of $v_i$.*

We provide a proof in Appendix B. The inherent mixed homophily-heterophily characteristics of fraud graphs underline the importance of utilizing node-specific adaptive filters. As different center nodes locate in diverse contexts within the graph, they each reflect distinct degrees of homophily or heterophily. This variability implies that a one-size-fits-all approach, using a universal graph filter across all nodes (Defferrard et al., 2016; Kipf & Welling, 2017; Tang et al., 2022), is not only suboptimal but could lead to inaccuracies in fraud detection. PMP ensures that the filters are tailored

Table 1: Experiment Results on Yelp and Amazon (40% training ratio).

| Method | Yelp | | | Amazon | | |
|---|---|---|---|---|---|---|
| | AUC | F1-Macro | G-Mean | AUC | F1-Macro | G-Mean |
| GCN | $59.83_{\pm0.49}$ | $56.20_{\pm0.67}$ | $43.65_{\pm2.62}$ | $83.69_{\pm1.25}$ | $64.86_{\pm6.94}$ | $57.18_{\pm19.51}$ |
| GAT | $57.15_{\pm0.29}$ | $48.79_{\pm2.30}$ | $16.59_{\pm7.89}$ | $81.02_{\pm1.79}$ | $64.64_{\pm3.87}$ | $66.75_{\pm13.45}$ |
| GraphSAGE | $89.38_{\pm0.19}$ | $75.46_{\pm0.81}$ | $73.07_{\pm4.79}$ | $93.16_{\pm0.87}$ | $88.26_{\pm0.62}$ | $83.46_{\pm1.49}$ |
| GPRGNN | $82.85_{\pm0.66}$ | $63.19_{\pm0.80}$ | $75.62_{\pm1.24}$ | $93.72_{\pm0.68}$ | $80.66_{\pm1.82}$ | $85.56_{\pm2.73}$ |
| FAGCN | $74.23_{\pm0.27}$ | $61.18_{\pm0.75}$ | $67.37_{\pm1.26}$ | $95.00_{\pm0.81}$ | $87.29_{\pm1.53}$ | $79.62_{\pm0.96}$ |
| Care-GNN | $76.19_{\pm2.92}$ | $63.32_{\pm0.94}$ | $67.91_{\pm3.59}$ | $90.67_{\pm1.49}$ | $86.39_{\pm1.66}$ | $70.52_{\pm0.21}$ |
| PC-GNN | $79.87_{\pm0.14}$ | $63.00_{\pm2.30}$ | $71.60_{\pm1.30}$ | $95.86_{\pm0.14}$ | $89.56_{\pm0.77}$ | $90.30_{\pm0.44}$ |
| H2-FDetector | $89.48_{\pm1.26}$ | $74.38_{\pm2.42}$ | $79.15_{\pm2.57}$ | $96.03_{\pm0.69}$ | $86.91_{\pm1.01}$ | $91.74_{\pm0.47}$ |
| BWGNN | $90.54_{\pm0.49}$ | $76.96_{\pm0.89}$ | $77.12_{\pm0.99}$ | $97.42_{\pm0.48}$ | $91.72_{\pm0.84}$ | $90.01_{\pm0.36}$ |
| GHRN | $90.57_{\pm0.36}$ | $77.54_{\pm1.02}$ | $74.21_{\pm1.57}$ | $97.07_{\pm0.73}$ | $\mathbf{92.36_{\pm0.97}}$ | $90.58_{\pm0.45}$ |
| GDN | $90.34_{\pm0.80}$ | $76.05_{\pm0.60}$ | $80.84_{\pm0.09}$ | $97.09_{\pm0.16}$ | $90.68_{\pm0.42}$ | $90.78_{\pm0.11}$ |
| **PMP** | $\mathbf{93.97_{\pm0.15}}$ | $\mathbf{81.96_{\pm0.56}}$ | $\mathbf{83.92_{\pm1.04}}$ | $\mathbf{97.57_{\pm0.12}}$ | $92.03_{\pm0.79}$ | $\mathbf{91.85_{\pm0.65}}$ |

Table 2: Experiment Results on T-Finance and T-Social (40% training ratio). OOM: out of memory; OOT: out of time (running time > 1 day).

| Method | T-Finance | | | T-Social | | |
|---|---|---|---|---|---|---|
| | AUC | F1-Macro | G-Mean | AUC | F1-Macro | G-Mean |
| GCN | $93.31_{\pm0.75}$ | $86.86_{\pm0.77}$ | $80.43_{\pm0.82}$ | $76.00_{\pm2.47}$ | $46.85_{\pm1.64}$ | $71.19_{\pm2.73}$ |
| GAT | $92.77_{\pm1.12}$ | $62.19_{\pm0.98}$ | $77.46_{\pm2.20}$ | $63.45_{\pm1.36}$ | $57.22_{\pm0.81}$ | $69.53_{\pm2.18}$ |
| GraphSAGE | $95.05_{\pm0.22}$ | $90.37_{\pm0.49}$ | $85.35_{\pm0.38}$ | $94.19_{\pm1.65}$ | $74.97_{\pm2.49}$ | $72.07_{\pm2.39}$ |
| GPRGNN | $54.82_{\pm0.78}$ | $56.25_{\pm0.27}$ | $32.96_{\pm0.25}$ | $82.79_{\pm1.21}$ | $49.23_{\pm0.87}$ | $50.03_{\pm7.62}$ |
| FAGCN | OOM | OOM | OOM | OOM | OOM | OOM |
| Care-GNN | $93.79_{\pm0.92}$ | $82.59_{\pm0.86}$ | $84.58_{\pm1.26}$ | $78.91_{\pm0.84}$ | $51.87_{\pm1.76}$ | $63.60_{\pm0.77}$ |
| PC-GNN | $92.09_{\pm0.58}$ | $55.81_{\pm0.34}$ | $81.96_{\pm0.42}$ | $88.98_{\pm0.66}$ | $44.13_{\pm1.07}$ | $75.07_{\pm0.80}$ |
| H2-FDetector | $94.99_{\pm0.68}$ | $74.21_{\pm0.70}$ | $85.91_{\pm0.69}$ | OOT | OOT | OOT |
| BWGNN | $95.84_{\pm0.46}$ | $88.66_{\pm0.72}$ | $85.16_{\pm1.66}$ | $94.72_{\pm1.88}$ | $84.06_{\pm2.89}$ | $81.51_{\pm4.08}$ |
| GHRN | $95.77_{\pm0.60}$ | $87.92_{\pm0.75}$ | $81.65_{\pm0.43}$ | $90.60_{\pm1.78}$ | $68.28_{\pm0.73}$ | $63.42_{\pm2.81}$ |
| GDN | $95.53_{\pm0.74}$ | $88.75_{\pm2.26}$ | $87.84_{\pm0.97}$ | $88.06_{\pm0.43}$ | $56.89_{\pm1.11}$ | $30.33_{\pm3.67}$ |
| **PMP** | $\mathbf{97.10_{\pm0.23}}$ | $\mathbf{91.90_{\pm0.50}}$ | $\mathbf{88.51_{\pm1.26}}$ | $\mathbf{99.62_{\pm0.07}}$ | $\mathbf{95.27_{\pm0.32}}$ | $\mathbf{92.97_{\pm0.64}}$ |

to the unique neighborhood characteristics of each node, enhancing both sensitivity and specificity in identifying fraud patterns. Fig. 2 shows an example of PMP on a $BA(500)$ graph. We can find that $g_{\text{fr}}^i(\Lambda)$ and $g_{\text{be}}^i(\Lambda)$ are two complementary filters that cover all frequencies in the spectral domain.

## 5 EXPERIMENTS

### 5.1 EXPERIMENTAL SETUP

**Datasets and Baselines.** We evaluate our approach using five datasets tailored for GFD: Yelp (Rayana & Akoglu, 2015), Amazon (McAuley & Leskovec, 2013), T-Finance, T-Social (Tang et al., 2022). Besides, we have evaluated our model on a large-scale real graph from our industry partner, Grab. The statistics of these datasets are provided in Table 4 of Appendix D. We compare our model against 11 state-of-the-art approaches, including *generic GNNs*: GCN (Kipf & Welling, 2017), GraphSAGE (Hamilton et al., 2017), and GAT (Veličković et al., 2018); *beyond homophily GNNs*: GPRGNN (Chien et al., 2021) and FAGCN (Bo et al., 2021); *fraud detection tailored GNNs*: Care-GNN (Dou et al., 2020), PC-GNN (Liu et al., 2021c), H2-FDetector (Shi et al., 2022), BWGNN (Tang et al., 2022), GHRN (Gao et al., 2023c), and GDN (Gao et al., 2023b).

**Metrics and Implementation Details.** We employ three commonly used metrics for imbalanced classification in deep learning evaluations: AUC, F1-Macro and G-Mean. We provide a detailed explanation of each metric in Appendix D.3. Following (Tang et al., 2022), we adopt the data splitting ratios of 40%:20%:40% for training, validation, and test set in the supervised scenario. In the semi-supervised scenario, the data splitting ratio is 1%:10%:89%. For consistency in our evaluations, each model underwent 10 trials with varied random seeds. We present the average performance

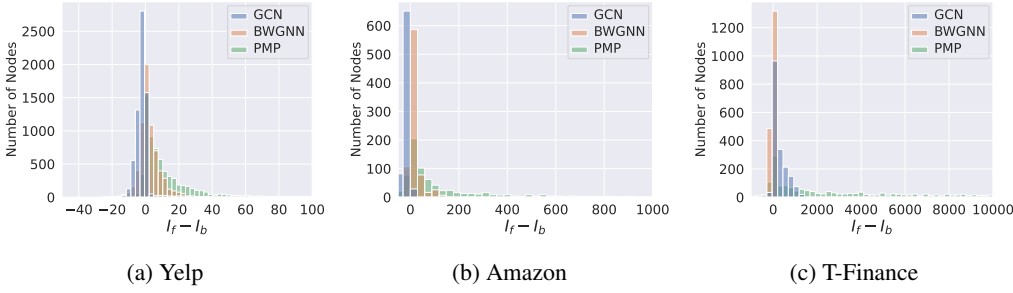

| (a) Yelp | (b) Amazon | (c) T-Finance |

Figure 4: Influence distribution.

and standard deviation for each model as benchmarks for comparison. We provide the detailed hyperparameter tuning strategies of baselines and hyperparameter setting of PMP in Appendix D.4.

## 5.2 Performance Comparison

For public datasets, results derived from the supervised setting can be found in Tables 1 and 2, while those from the semi-supervised setting are detailed in Tables 6 and 7 of Appendix E.1. For the Grab dataset, results are shown in Appendix E.2. The results demonstrate that PMP consistently surpasses baseline performances across almost all datasets and metrics. One explanation for its enhanced performance over generic GNNs—where the learnable weights of the aggregation function are uniformly applied across all neighbors—is that PMP encodes class-specific discriminative information into the model parameters.

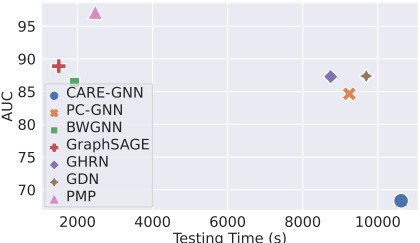

Figure 3: AUC *vs.* testing time on T-Social.

This is achieved by distinguishing neighbors of distinct classes during the message passing phase. Besides, the adaptive modulation between $\mathbf{W}_{\text{fr}}$ and $\mathbf{W}_{\text{be}}$ allows nodes to judiciously calibrate the information flow from distinct classes of neighbors. By segregating the processing of neighbors based on their labels, our model can adaptively emphasize and give importance to rare patterns, which is vital in imbalanced and heterophilic scenarios. Among generic GNNs, GraphSAGE exhibits better performance due to its separation between ego- and neighbor embeddings, which is beneficial when learning under heterophily, and our model also inherits such design.

In comparison to the six GNNs tailored for GFD, our model also demonstrates a markedly superior performance. For instance, on Yelp, our model shows improvements of 3.4% in AUC, 4.42% in F1-Macro, and 3.08% in G-Mean. Interestingly, our evaluation reveals that GraphSAGE, despite its fundamental design, serves as a potent baseline, even surpassing some models specifically crafted for GFD. Many of these specialized baselines incorporate intricate preprocessing steps rooted in feature engineering (e.g., GHRN) or employ learnable edge reweighting/sampling techniques (e.g., H2-FDetector) for graph augmentation. Such findings suggest that elaborate manipulations to tailor the graph structure for the model might be superfluous. Instead, adapting the model to align better with the inherent graph characteristics could prove to be a more effective strategy. Our model presented in Eq. (3) can be seen as a variant of GraphSAGE. Notably, relative to GraphSAGE, our model exhibits substantial performance enhancements, underscoring the efficacy of the proposed partitioning message passing strategy. In Fig. 3, we show that our model achieves the optimal trade-off between inference speed and effectiveness in comparison to baselines. PMP follows the iterative aggregation framework of GNNs, thus the number of layers and the dimension of hidden layers are two core parameters that determine the receptive field and representation power of the model. Thus, we analyze the sensitivity of the model to these two parameters in Appendix D.5.

## 5.3 How does PMP Solve the Problems of Heterophily and Label Imbalance?

To answer this question, we investigate the influence of neighbors with different labels on the center fraud nodes. Given a center fraud node $v_i$, we measure the influence of its fraud neighbors on the final representation of $v_i$ as $I_f(v_i) = \sum_{v_j \in \mathcal{N}_{\text{fr}}(v_i)} \frac{\partial Z_i}{\partial X_j}$, where $Z_i = H_i^{(L)}$ is the output representation of the last layer. Correspondingly, the influence of benign neighbors is captured by

Table 3: Ablation study

| Method | Yelp | | Amazon | | T-Finance | |
|---|---|---|---|---|---|---|
| | AUC | F1-Macro | AUC | F1-Macro | AUC | F1-Macro |
| GraphSAGE | 89.38 | 75.46 | 93.16 | 88.26 | 95.05 | 90.37 |
| +partition | 93.15 | 78.02 | 96.33 | 89.64 | 95.92 | 91.43 |
| ++adaptive combination | 93.51 | 79.93 | **97.61** | 91.29 | 96.86 | 91.31 |
| +++root-specific weights (full PMP) | **93.97** | **81.96** | 97.57 | **92.03** | **97.10** | **91.90** |

$I_b(v_i) = \sum_{v_j \in \mathcal{N}_{\text{be}}(v_i)} \frac{\partial Z_i}{\partial X_j}$. Then, the larger $I_f(v_i) - I_b(v_i)$, the larger influence of homophilic neighbors on the center fraud node, which indicates that the model is robust, effectively mitigating the issues of label imbalance and heterophily. In Fig. 4, we show the distribution of $I_f - I_b$ with respect to all fraud nodes. It is evident that, in comparison to GCN and BWGNN, the representations learned through PMP exhibit a higher $I_f - I_b$ for a greater number of fraud nodes. This metric underscores PMP's efficacy in enhancing the impact of homophilic (fraud) neighbors on the center fraud node, despite them being in the minority, on the center node, thereby showcasing its superior ability to resist the challenges posed by label imbalance and heterophily.

## 5.4 ABLATION STUDY

As shown in Eq. (3), the main difference between PMP and GraphSAGE rests in their aggregation methods: while GraphSAGE employs a uniform aggregation with shared feature transformations for all neighbors, PMP distinctly partitions message passing, applying varied feature transformations contingent upon neighbor labels. The results in Section 5.2 show our model consistently outperforms GraphSAGE across all datasets and evaluation metrics, which demonstrates that such a simple design can significantly enhance GNN performance in GFD.

Additionally, we delve deeper to validate two pivotal components of our model: the adaptive blending of unlabeled neighbors as a weighted fusion of labeled data, as shown in Eq. (4), and the root-specific weight matrices in Eq. (5). Table 3 shows the results, where we employ GraphSAGE as the benchmark model, given its conceptual proximity to our proposal, to methodically illustrate the incremental benefits introduced by our design choices. "+partition" denotes adopting distinct weight matrices for labeled fraud benign neighbors, while for unlabeled neighbors, a separate, independent weight matrix is employed. Subsequently, "++adaptive combination" denotes that the weight matrix for unlabeled neighbors is treated as an adaptive combination of the weights of labeled neighbors as introduced in Eq. (4). "+++root-specific weights" implies that the weight matrices for fraud and benign neighbors are dynamically generated as functions of the center node as presented in Eq. (5).

From our evaluations, it is evident that GraphSAGE lags behind in performance across all metrics. This underscores the inherent limitation of uniformly aggregating information from all neighbors during the message passing process. Such an approach, while general and versatile, might miss out on capturing the relationships and contextual dependencies intrinsic to GFD. The "+partition" contributes to major improvements, which affirms that the core feature of our model, distinguishing between different classes of neighbors during message passing, stands as a pivotal mechanism for enhancing GNNs' efficacy in GFD. Besides, the subsequent extensions, "adaptive combination" and "root-specific weights", also contribute positively to the overall performance. Specifically, the "++adaptive combination", building upon the "+partition", brings incremental improvements across all datasets. The "+++root-specific weights", further extending the preceding component, enhances performance on most metrics in most datasets. Combined, these two components contribute to a cumulative improvement of more than 1% over the "+partition" baseline.

## 6 CONCLUSIONS

This work presents a simple yet effective GNN framework, PMP, for fraud detection task. We propose that the key of applying GNNs on fraud detection is to distinguish neighbors with different labels during message passing. With this insight, neighbor aggregation is partitioned based on labels using distinct node-specific aggregation functions. In this way, the center node can adaptively adjust the information propagated from its homophilic and heterophilic neighbors, thus avoiding model overfitting the majority class nodes (i.e., benign), and alleviating the problem of heterophily. The theoretical analysis of PMP demonstrates that PMP learns an adaptive spectral filter for each node separately. Extensive experiments show that our model achieves new state-of-the-art results on several public GFD datasets, verifying the power of partitioning message passing for GFD.

ACKNOWLEDGMENTS

This research is supported by the National Research Foundation, Singapore under its AI Singapore Programme (AISG Award No: AISG2-TC-2021-002); Shenzhen Baisc Research Fund under grant JCYJ20200109142217397.

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

# A  ALGORITHMIC DETAILS

---

**Algorithm 1** PMP forward propatation

---

**Input:** Fraud graph $\mathcal{G} = (\mathcal{V}, E_r, \mathbf{X}, \mathcal{Y})$; Depth $L$; Batch size $B$;
**Output:** Logits $\mathbf{Z} \in \mathbb{R}^N$

1: **for** $l \in \{1, \cdots, L\}$ **do**
2:    **for** each $batch \subseteq \mathcal{G}$ of size $B$ **do**
3:       **for** $v_i \in batch$ **do**
4:          $\widehat{\mathbf{h}}_i^{(l)} \leftarrow f_{\text{self}}^{(l-1)}(\mathbf{h}_i^{(l-1)})$
5:          $\alpha_i^{(l-1)} \leftarrow \Phi(\mathbf{h}_i^{(l-1)})$
6:          $\mathbf{W}_{\text{fr},i}^{(l-1)} \leftarrow \Psi_{\text{fr}}(\mathbf{h}_i^{(l-1)})$; $\mathbf{W}_{\text{be},i}^{(l-1)} = \Psi_{\text{be}}(\mathbf{h}_i^{(l-1)})$; $\mathbf{W}_{\text{un},i}^{(l-1)} = \alpha_i^{(l-1)}\mathbf{W}_{\text{fr}}^{(l-1)} + (1 - \alpha_i^{(l-1)})\mathbf{W}_{\text{be},i}^{(l-1)}$ // Generate weight matrices for $v_i$
7:          $\mathbf{a}_i^{(l)} = f_{\text{fr},i}^{(l-1)}\left(\mathbf{h}_j^{(l-1)}|v_j \in \mathcal{N}_{\text{fr}}(v_i)\right) + f_{\text{be},i}^{(l-1)}\left(\mathbf{h}_j^{(l-1)}|v_j \in \mathcal{N}_{\text{be}}(v_i)\right) + f_{\text{un},i}^{(l-1)}\left(\mathbf{h}_j^{(l-1)}|v_j \notin \mathcal{N}_{\text{be}}(v_i) \cup \mathcal{N}_{\text{fr}}(v_i)\right)$ // $f_{\text{fr},i}^{(l-1)}$, $f_{\text{be},i}^{(l-1)}$ and $f_{\text{un},i}^{(l-1)}$ are parameterized by $\mathbf{W}_{\text{fr},i}^{(l-1)}$, $\mathbf{W}_{\text{be},i}^{(l-1)}$ and $\mathbf{W}_{\text{un},i}^{(l-1)}$ respectively.
8:          $\mathbf{h}_i^{(l)} = \widehat{\mathbf{h}}_i^{(l)} + \mathbf{a}_i^{(l)}$
9:       **end for**
10:    **end for**
11: **end for**
12: $\widetilde{\mathbf{H}} = \text{MLP}(\mathbf{H}^{(L)}) \in \mathbb{R}^{N \times 1}$
13: $\mathbf{Z}_i \leftarrow \text{Sigmoid}(\widetilde{\mathbf{H}})$
14: $\mathcal{L} = \sum_i \left(y_i \log(\mathbf{Z}_i) + (1 - y_i) \log(1 - \mathbf{Z}_i)\right)$ // Cross-entropy loss

---

**Time Complexity**    The time complexity of an $L$-layer PMP propagation is $\mathcal{O}(LNd + L|E|d^2)$. For the adaptive combination of unlabeled neighbors, a single-layer MLP is employed, resulting in a complexity of $\mathcal{O}(Nd)$. Additionally, generating the root-specific weight matrix carries a time complexity of $\mathcal{O}(Nd^2)$. Thus, the aggregate time complexity of the PMP can be summarized as $\mathcal{O}(LNd + (L|E| + N)d^2)$.

# B  PROOF OF THEOREM 1

*Proof.* Assuming the node indices are fixed, let $\mathbf{F}$, $\mathbf{B}$ be diagonal label mask matrices, which mask benign nodes and fraud nodes respectively with 0 in the main diagonal elements. Specifically, for a node $v_i$ labeled 1 (fraud) in the training set, we assign $\mathbf{F}_{ii} = 1$, otherwise 0. Similarly, for training benign nodes, $\mathbf{B}_{ii} = 1$ otherwise 0. Then the mask matrix of unlabeled nodes is thus $\mathbf{I} - \mathbf{F} - \mathbf{B}$. For the sake of simplicity, we analyze the first layer of PMP as an example and omit superscripts. Specifically, the feature transformation step of Eq. (3) can be reformulated as:

$$\begin{aligned}
&\mathbf{FXW}_{\text{fr}} + \mathbf{BXW}_{\text{be}} + (\mathbf{I} - \mathbf{F} - \mathbf{B})\mathbf{X}\left(\alpha_i \mathbf{W}_{\text{fr}} + (1 - \alpha_i)\mathbf{W}_{\text{be}}\right) \\
&= (\mathbf{F} + \alpha_i\mathbf{I} - \alpha_i\mathbf{F} - \alpha_i\mathbf{B})\mathbf{XW}_{\text{fr}} + (\mathbf{B} + (1 - \alpha_i)\mathbf{I} - (1 - \alpha_i)\mathbf{B} - (1 - \alpha_i)\mathbf{B})\mathbf{XW}_{\text{be}}.
\end{aligned} \quad (8)$$

Let $\mathbf{K}(v_i) = \mathbf{F} + \alpha_i\mathbf{I} - \alpha_i\mathbf{F} - \alpha_i\mathbf{B}$, Eq. (8) can be rewritten as:

$$\mathbf{K}(v_i)\mathbf{XW}_{\text{fr}} + (\mathbf{I} - \mathbf{K}(v_i))\mathbf{XW}_{\text{be}}$$

$$\text{where} \quad \mathbf{K}(v_i)[j,j] = \begin{cases} 1 & v_j \in \mathcal{N}_{\text{fr}}(v_i) \\ 0 & v_j \in \mathcal{N}_{\text{be}}(v_i) \\ \alpha_i & \text{otherwise} \end{cases}. \quad (9)$$

To facilitate the spectral analysis, we align the definition of the graph convolution with GCNs (Kipf & Welling, 2017), using the normalized adjacency matrix, $\mathbf{D}^{-1/2}\mathbf{AD}^{-1/2} = \mathbf{I} - \mathbf{L}$. Note that although our model utilizes summation as the convolution method, this doesn't alter its inherent

spectral properties (Dong et al., 2021; Zhu et al., 2021). Since the matrix $\mathbf{K}(\cdot)$ is node-specific, for $v_i$, it induces a separate message passing matrix form:

$$
\begin{aligned}
\mathbf{H}(v_i) &= (\mathbf{I} - \mathbf{L})\mathbf{K}(v_i)\mathbf{X}\mathbf{W}_{\text{fr}} + (\mathbf{I} - \mathbf{L})(\mathbf{I} - \mathbf{K}(v_i))\mathbf{X}\mathbf{W}_{\text{be}} \\
&= \mathbf{U}g_{\text{fr}}^i(\Lambda)\mathbf{U}^\top \mathbf{X}\mathbf{W}_{\text{fr}} + \mathbf{U}g_{\text{be}}^i(\Lambda)\mathbf{U}^\top \mathbf{X}\mathbf{W}_{\text{be}},
\end{aligned}
\tag{10}
$$

where the node-specific convolution filters $g_{\text{fr}}^i(\Lambda)$ and $g_{\text{be}}^i(\Lambda)$ can be derived as Eq. (7). $\mathbf{H}(v_i) \in \mathbb{R}^{N \times d'}$ is a representation matrix induced by the convolution kernel functions associated with $v_i$, thus its $i$-th row $\mathbf{H}(v_i)[i,:]$ gives the PMP representation for $v_i$. For another node $v_j$, it yields a corresponding PMP representation matrix $\mathbf{H}(v_j)$ whose $j$-th row is the representation of $v_j$. Besides, we can find that the spectral filters depend on the node indices and the label distribution. For the frequency $\lambda_j$ with its index $j$ satisfying $v_j \notin \mathcal{N}_{\text{fr}}(v_i) \cup \mathcal{N}_{\text{be}}(v_i)$, then its response $g^i(\lambda_j)$ is node-adaptive given that $\alpha_i$ is inherently a function of $v_i$. Since in real-world networks, where most nodes remain unlabeled, we can conclude that PMP is an adaptive node-specific spectral convolution. $\square$

## C    RELATION OF PMP TO ACM

The basic idea of Adaptive Channel Mixture (ACM) (Luan et al., 2021) is to attribute negative weights to heterophilic neighbors through a spectral synthesis of low-pass and high-pass filters. This technique fundamentally differs from our PMP. It is because the use of negative weights in ACM is indicative of removing (or subtracting) the heterophilic components from the homophilic neighbors. This can be interpreted as a form of information "forgetting" where heterophilic features are de-emphasized in favor of homophilic features. It is different from PMP which preserves and utilizes all available information. From the perspective of methodology, ACM is rooted in spectral graph theory, while PMP is spatially oriented. This difference is not just theoretical but has practical implications. The spectral nature of ACM inherently influences its scalability, particularly in the context of large graphs. Typically, spectral-based models, including ACM, require the entire graph as input, which can pose challenges for mini-batch training and thereby limit scalability. In contrast, our PMP model, with its spatial-based framework, inherently supports mini-batch training and is thus more adaptable to large-scale graphs. Furthermore, considering the application of ACM in GFD tasks, its formulation, with removed nonlinearity for simplicity's sake, can be expressed as $H_{\mathbf{ACM}}^{(l+1)} = \alpha F_L \begin{bmatrix} H_{fr}^{(l)} \\ H_{be}^{(l)} \\ H_{un}^{(l)} \end{bmatrix} W_1^{(l)} + \beta F_H \begin{bmatrix} H_{fr}^{(l)} \\ H_{be}^{(l)} \\ H_{un}^{(l)} \end{bmatrix} W_2^{(l)}$ where $F_L$ and $F_H$ are distinct spectral filters. We can observe that the trainable weight matrix $W_L^{(l)}$ and $W_H^{(l)}$ are both shared across all nodes. In contrast, our PMP model, when represented in matrix form, can be rewritten as $H_{\mathbf{PMP}}^{(l+1)} = F \begin{bmatrix} H_{fr}^{(l)} W_1 \\ H_{be}^{(l)} W_2 \\ H_{un}^{(l)}(\alpha W_1 + (1-\alpha)W_2) \end{bmatrix}$ where $F$ is the normalized adjacency matrix. Unlike ACM, trainable weight matrices in PMP, $W_1$ and $W_2$, are specifically applied to nodes with different labels, where $W_1$ and $W_2$ capture the information of fraud nodes and benign nodes respectively, reflecting a more label-aware and adaptive approach to message passing in the context of GFD.

## D    EXPERIMENTAL DETAILS

### D.1    DATASETS

Table 4 summarizes the dataset statistics, including the number of nodes, edges, and relations, the proportion of fraud nodes, feature dimension, and the homophily score. The imbalance-aware homophily score (Lim et al., 2021) is defined as:

$$
\hat{\eta} = \frac{1}{C-1} \sum_{k=0}^{C-1} \left[ \eta_k - \frac{|C_k|}{N} \right]_+
\tag{11}
$$

where $[\cdot]_+ = \max(\cdot, 0)$. $C$, representing the number of classes, is set to 2 for GFD. $C_k$ denotes the set of nodes in class $k$, with $k = 0$ corresponding to benign nodes and $k = 1$ to fraud nodes. $\eta_k$ is the

class-wise homophily metric:

$$\eta_k = \frac{\sum_{v_i \in C_k} |\mathcal{N}_k(v_i)|}{\sum_{v_i \in C_k} |\mathcal{N}(v_i)|} \tag{12}$$

where $\mathcal{N}_k(v_i)|$ is the neighbors of $v_i$ with label $k$. $\hat{\eta} \in [0, 1]$ where $\hat{\eta} = 1$ corresponds to a fully homophilic graph, while $\hat{\eta} < 0.5$ indicates a highly heterophilic graph.

We compare the fraud detection performance of different baselines on YelpChi, Amazon, T-Finance, and T-Social. Among them, 1) YelpChi comprises both filtered (spam) and recommended (legitimate) reviews of hotels and restaurants, as collected by Yelp.com. This dataset has three types of relations: R-U-R, representing reviews by the same user; R-S-R, for reviews under the same product receiving identical star ratings; and R-T-R, which groups reviews for the same product posted within the same month. 2) The Amazon dataset collects reviews from the musical instruments category on Amazon. It also has three relations, where the U-P-U relationship links users who have reviewed the same product; the U-S-U relationship connects users who assigned identical ratings within a week; the U-V-U relationship bridges users showcasing the top 5% of mutual review text similarities measured by TF-IDF metrics. 3) T-Finance is a single-relational fraud graph, capturing anomalous accounts within transaction networks. In this dataset, nodes represent anonymized accounts, with attributes including registration days, login activities, and interaction frequencies. Edges mean the existence of transaction records between two accounts. Nodes are annotated as anomalies by human experts if they exhibit behaviors characteristic of fraud, money laundering, or online gambling. 4) T-Social is designed to identify anomalous accounts within social networks. two nodes are interconnected if they sustain a friendship for over three months. 5) The industrial graph includes the transaction of a month in a leading super app. Due to the anonymity and privacy policy, we exclude the exact details of the graph but roughly the graph includes over 1 million nodes and over 10 million edges.

Table 4: Summary of dataset statistics.

| Dataset | # Nodes | # Edges | # Relations | Frauds (%) | # Features | $\hat{\eta}$ |
|---|---|---|---|---|---|---|
| YelpChi | 45,954 | 3,846,979 | 3 | 14.53% | 32 | 0.0538 |
| Amazon | 11,944 | 4,398,392 | 3 | 6.87% | 25 | 0.0512 |
| T-Finance | 39,357 | 21,222,543 | 1 | 4.58% | 10 | 0.4363 |
| T-Social | 5,781,065 | 73,105,508 | 1 | 3.01% | 10 | 0.1003 |
| Industrial graph | ∼1M | ∼10M | 1 | 0.54% | 17 | - |

## D.2 LABEL DISTRIBUTION

In Fig. 5, we show the label distributions of the labeled neighborhoods of training nodes. We use $\frac{|\mathcal{N}_{fr}|}{|\mathcal{N}_{be}|}$ to represent the number of fraud neighbors relative to benign neighbors for a central node. $\frac{|\mathcal{N}_{fr}|}{|\mathcal{N}_{be}|} < 1$ indicates that the number of labeled fraud neighbors is fewer than benign neighbors. Specifically, if $\frac{|\mathcal{N}_{fr}|}{|\mathcal{N}_{be}|} < 0.5$, it indicates that the central node has far fewer fraud neighbors compared to benign ones, denoting a highly imbalanced neighborhood. The height of each bar means the number of center training nodes under a certain $\frac{|\mathcal{N}_{fr}|}{|\mathcal{N}_{be}|}$. Our results show that commonly used datasets such as Yelp, Amazon, and T-Finance suffer from significant label imbalance in the neighborhoods, because the neighborhoods' label distribution is long-tail, i.e., the number of fraud neighbors is far less than benign ones. It requires the model can enhance the influence from the minority class neighbors on the center nodes during neighbor aggregation. Combining with the empirical results in Section 5.3, our PMP method effectively achieves this, enhancing the influence from minority class neighbors more efficiently than traditional GNN approaches.

## D.3 METRICS

AUC is the area under the ROC curve, and it provides an aggregate measure of performance across all possible classification thresholds, reflecting the model's ability to distinguish between positive and negative classes. F1-Macro computes the F1 score for each class independently and then takes the average. Lastly, the G-Mean, or geometric mean, calculates the square root of the product of the sensitivity and specificity, offering an insight into the balance between true positive rate and

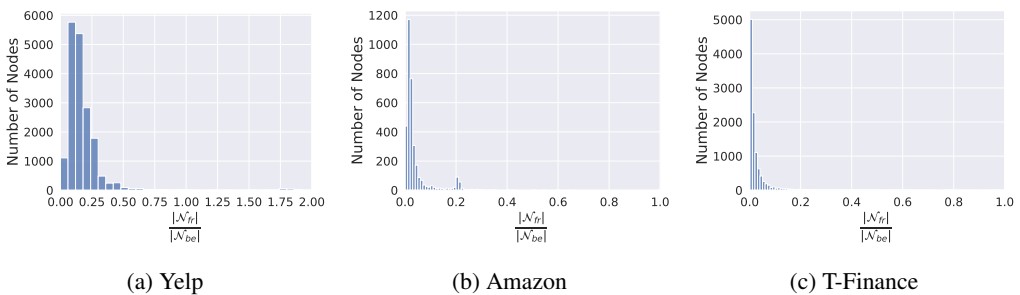

Figure 5: Label distributions of the labeled neighborhoods of the training nodes.

true negative rate performance. Higher values of these metrics signify superior performance of the methods.

## D.4 HYPERPARAMETER SETTINGS

To mitigate bias, we individually fine-tune the hyperparameters of each model for every benchmark and report the best performance on the validation set. For each model, we explored the following searching ranges for general hyperparameters: learning rate $lr \in \{0.01, 0.005, 0.001\}$, weight decay $wd \in \{0, 5e-5, 1e-4\}$, dropout $do \in \{0, 0.2, 0.3, 0.4, 0.5, 0.6, 0.7, 0.8\}$, hidden dimension $d' \in \{32, 64, 128, 256, 512\}$. For spatial-based models, batch size is an important hyperparameter that highly depends on the graph size. Specifically, for Yelp, Amazon, and T-Finance, batch size $bs \in \{64, 128, 256, 512, 1024\}$. For the large-scale graph T-Social, $bs \in \{2^{17}, 2^{18}, 2^{19}\}$. For model-specific hyperparameters, we also carefully calibrate parameters in accordance with varying datasets and training sizes. Here we provide the optimal hyperparameters of PMP in Table 5.

Table 5: Hyperparameter settings for PMP.

| Dataset | $lr$ | $wd$ | $do$ | $bs$ | $L$ | $d'$ |
|---|---|---|---|---|---|---|
| Yelp | 0.01 | 0 | 0 | 512 | 1 | 256 |
| Amazon | 0.01 | 0 | 0.6 | 128 | 1 | 128 |
| T-Finance | 0.01 | 0 | 0.4 | 256 | 1 | 64 |
| T-Social | 0.001 | 0 | 0 | $2^{17}$ | 1 | 128 |

## D.5 PARAMETER STUDY

We investigate the sensitivity in relation to the key hyperparameters in our model: the number of layers $L$ and the hidden dimension $d'$. For each dataset, we vary $L$ over the range $[1, 2, 3]$, and $d'$ over $[32, 64, 128, 256, 512]$. We employed a grid search methodology to test combinations of hyperparameters. As illustrated in Fig. 6, PMP consistently performs optimally with a single layer, and more layers bring about a continuous decrease in effect. This phenomenon can be attributed to the issue of oversmoothing. We find that all these datasets have a high average degree. Specifically, the average degree of Yelp, Amazon and T-Finance are 167, 740, and 1078, respectively, and we further investigate the number of nodes within the first two hops of neighborhoods. Eliminating duplication, the average number of nodes in the first two-hop neighborhoods across all nodes stands at 1229 for Yelp, 11338 for Amazon, and 24480 for T-Finance. The remarkable density of these benchmark datasets implies that even a two-layer GNN might aggregate an overwhelming quantum of information from the global, thereby obfuscating the essential local features. Additionally, Furthermore, our findings reveal that the optimal value of $d'$ varies across datasets. This calls for meticulous tuning to ascertain peak performance. A potential rationale for this sensitivity lies in the substantial variability in the dimensions of input node features across different datasets.

## D.6 COMPUTING RESOURCES

For all experiments, we use a single NVIDIA A100 GPU with 80GB GPU memory.

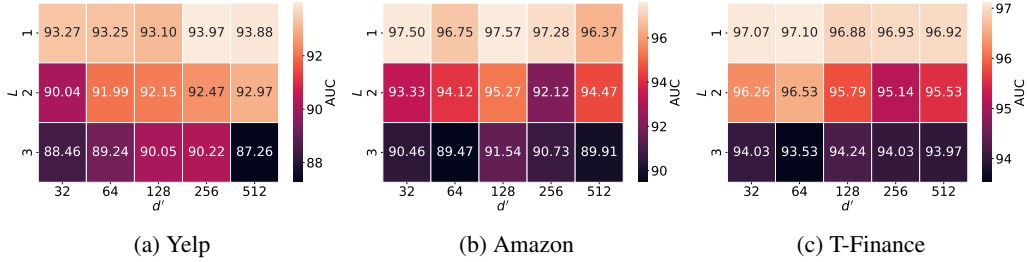

Figure 6: Sensitivity study of different hyperparameter combinations.

# E MORE EXPERIMENTAL RESULTS

## E.1 SEMI-SUPERVISED FRAUD DETECTION ON PUBLIC DATASETS

In Table 6 and Table 7, we show the performance of fraud detection under a semi-supervised setting (1% training ratio).

Table 6: Experiment Results on Yelp and Amazon (1% training ratio).

| Method | Yelp | | | Amazon | | |
|---|---|---|---|---|---|---|
| | AUC | F1-Macro | G-Mean | AUC | F1-Macro | G-Mean |
| GCN | $54.06_{\pm 0.72}$ | $52.48_{\pm 0.50}$ | $46.53_{\pm 0.98}$ | $82.85_{\pm 0.71}$ | $67.93_{\pm 1.42}$ | $58.12_{\pm 1.97}$ |
| GAT | $50.95_{\pm 1.39}$ | $50.27_{\pm 2.31}$ | $26.83_{\pm 8.94}$ | $73.45_{\pm 1.26}$ | $60.84_{\pm 2.47}$ | $42.24_{\pm 5.73}$ |
| GraphSAGE | $82.59_{\pm 0.26}$ | $64.21_{\pm 9.12}$ | $57.09_{\pm 9.56}$ | $87.50_{\pm 0.66}$ | $77.88_{\pm 2.53}$ | $70.00_{\pm 2.26}$ |
| GPRGNN | $54.19_{\pm 2.95}$ | $46.07_{\pm 1.38}$ | $8.29_{\pm 2.63}$ | $80.77_{\pm 0.31}$ | $51.14_{\pm 2.38}$ | $19.93_{\pm 4.69}$ |
| FAGCN | $70.92_{\pm 0.92}$ | $46.27_{\pm 0.28}$ | $5.90_{\pm 3.17}$ | $92.07_{\pm 1.51}$ | $84.82_{\pm 0.73}$ | $80.13_{\pm 1.25}$ |
| Care-GNN | $73.96_{\pm 0.13}$ | $61.25_{\pm 0.34}$ | $63.87_{\pm 0.20}$ | $88.30_{\pm 0.60}$ | $69.24_{\pm 0.27}$ | $78.17_{\pm 0.26}$ |
| PC-GNN | $75.35_{\pm 0.15}$ | $55.05_{\pm 0.21}$ | $68.05_{\pm 0.25}$ | $91.73_{\pm 0.52}$ | $87.58_{\pm 0.20}$ | $80.35_{\pm 1.68}$ |
| H2-FDetector | $74.19_{\pm 0.52}$ | $57.42_{\pm 0.45}$ | $67.92_{\pm 0.21}$ | $83.26_{\pm 0.17}$ | $67.60_{\pm 0.31}$ | $55.25_{\pm 0.36}$ |
| BWGNN | $79.31_{\pm 0.25}$ | $66.59_{\pm 0.16}$ | $66.12_{\pm 0.36}$ | $88.37_{\pm 0.77}$ | $86.50_{\pm 0.59}$ | $83.35_{\pm 0.46}$ |
| GHRN | $76.76_{\pm 0.37}$ | $64.30_{\pm 0.61}$ | $61.73_{\pm 1.03}$ | $90.27_{\pm 0.30}$ | $\mathbf{89.16}_{\pm 0.89}$ | $83.68_{\pm 2.22}$ |
| GDN | $72.31_{\pm 0.36}$ | $58.28_{\pm 2.13}$ | $53.07_{\pm 1.00}$ | $86.38_{\pm 0.29}$ | $76.20_{\pm 0.08}$ | $83.19_{\pm 0.59}$ |
| **PMP** | $\mathbf{83.94}_{\pm 0.37}$ | $\mathbf{68.60}_{\pm 0.76}$ | $\mathbf{72.39}_{\pm 0.42}$ | $\mathbf{91.82}_{\pm 0.74}$ | $87.72_{\pm 1.15}$ | $\mathbf{83.77}_{\pm 0.66}$ |

Table 7: Experiment Results on T-Finance and T-Social (1% training ratio)

| Method | T-Finance | | | T-Social | | |
|---|---|---|---|---|---|---|
| | AUC | F1-Macro | G-Mean | AUC | F1-Macro | G-Mean |
| GCN | $74.37_{\pm 0.73}$ | $57.51_{\pm 1.62}$ | $72.58_{\pm 0.90}$ | $59.61_{\pm 1.37}$ | $52.34_{\pm 0.31}$ | $52.35_{\pm 3.93}$ |
| GAT | $83.38_{\pm 0.67}$ | $67.18_{\pm 0.3}$ | $59.71_{\pm 0.92}$ | $68.55_{\pm 1.46}$ | $47.03_{\pm 1.27}$ | $71.26_{\pm 0.86}$ |
| GraphSAGE | $92.53_{\pm 0.51}$ | $85.09_{\pm 0.64}$ | $82.86_{\pm 1.73}$ | $88.89_{\pm 0.77}$ | $49.24_{\pm 0.83}$ | $22.76_{\pm 7.66}$ |
| GPRGNN | $89.57_{\pm 0.65}$ | $80.47_{\pm 1.19}$ | $69.34_{\pm 1.95}$ | $74.30_{\pm 2.57}$ | $49.24_{\pm 0.84}$ | $2.72_{\pm 1.10}$ |
| FAGCN | OOM | OOM | OOM | OOM | OOM | OOM |
| Care-GNN | $90.73_{\pm 0.24}$ | $76.25_{\pm 0.29}$ | $74.38_{\pm 0.63}$ | $68.32_{\pm 1.73}$ | $53.31_{\pm 1.52}$ | $63.75_{\pm 0.60}$ |
| PC-GNN | $92.29_{\pm 0.17}$ | $76.17_{\pm 0.16}$ | $81.93_{\pm 0.73}$ | $84.66_{\pm 1.57}$ | $49.38_{\pm 0.94}$ | $57.06_{\pm 2.65}$ |
| H2-FDetector | $\mathbf{94.37}_{\pm 1.66}$ | $83.44_{\pm 2.53}$ | $79.32_{\pm 2.02}$ | OOT | OOT | OOT |
| BWGNN | $89.54_{\pm 5.03}$ | $85.17_{\pm 2.81}$ | $78.99_{\pm 5.12}$ | $86.40_{\pm 14.62}$ | $77.35_{\pm 12.89}$ | $72.17_{\pm 16.48}$ |
| GHRN | $89.18_{\pm 2.31}$ | $73.20_{\pm 1.54}$ | $70.08_{\pm 1.65}$ | $87.28_{\pm 0.29}$ | $65.37_{\pm 0.33}$ | $57.93_{\pm 0.17}$ |
| GDN | $93.32_{\pm 0.69}$ | $87.26_{\pm 1.77}$ | $83.04_{\pm 1.04}$ | $87.37_{\pm 0.80}$ | $56.48_{\pm 2.85}$ | $29.34_{\pm 3.34}$ |
| **PMP** | $93.78_{\pm 0.19}$ | $\mathbf{88.89}_{\pm 0.37}$ | $\mathbf{84.91}_{\pm 2.57}$ | $\mathbf{97.14}_{\pm 0.75}$ | $\mathbf{84.58}_{\pm 0.30}$ | $\mathbf{83.16}_{\pm 0.64}$ |

## E.2 EXPERIMENTAL RESULTS ON GRAB

In Table 8 and Table 9, we show the fraud detection performance on the Grab dataset under supervised and semi-supervised settings.

## E.3 COMPARISON WITH R-GCN

As an extension of the comparative analysis presented in Fig. 4, we include R-GCN due to its ability to handle multiple types of relationships within a graph. R-GCN (Schlichtkrull et al., 2018) extends

Table 8: Experiment Results on Grab (40% training ratio)

| Method | Grab | | |
|---|---|---|---|
| | AUC | F1-Macro | G-Mean |
| Care-GNN | $99.58_{\pm 0.01}$ | $68.87_{\pm 0.21}$ | $98.31_{\pm 0.03}$ |
| PC-GNN | $98.97_{\pm 0.02}$ | $66.32_{\pm 1.15}$ | $98.08_{\pm 0.14}$ |
| H2-FDetector | OOT | OOT | OOT |
| BWGNN | $99.79_{\pm 0.03}$ | $80.63_{\pm 0.47}$ | $99.32_{\pm 0.01}$ |
| GHRN | $99.71_{\pm 0.01}$ | $76.65_{\pm 1.46}$ | $99.11_{\pm 0.01}$ |
| GDN | $99.73_{\pm 0.04}$ | $80.83_{\pm 1.47}$ | $\mathbf{99.64_{\pm 0.02}}$ |
| **PMP** | $\mathbf{99.82_{\pm 0.02}}$ | $\mathbf{82.42_{\pm 0.90}}$ | $99.63_{\pm 0.08}$ |

Table 9: Experiment Results on Grab (1% training ratio)

| Method | Grab | | |
|---|---|---|---|
| | AUC | F1-Macro | G-Mean |
| Care-GNN | $99.56_{\pm 0.03}$ | $71.73_{\pm 0.22}$ | $98.65_{\pm 0.02}$ |
| PC-GNN | $96.35_{\pm 0.24}$ | $71.51_{\pm 2.33}$ | $98.98_{\pm 0.15}$ |
| H2-FDetector | OOT | OOT | OOT |
| BWGNN | $98.78_{\pm 0.12}$ | $\mathbf{81.53_{\pm 0.79}}$ | $99.40_{\pm 0.02}$ |
| BHomo-GHRN | $99.53_{\pm 0.02}$ | $69.78_{\pm 1.07}$ | $98.42_{\pm 0.26}$ |
| GDN | $\mathbf{99.72_{\pm 0.02}}$ | $79.33_{\pm 0.46}$ | $99.27_{\pm 0.43}$ |
| **PMP** | $\mathbf{99.72_{\pm 0.00}}$ | $79.73_{\pm 0.06}$ | $\mathbf{99.54_{\pm 0.04}}$ |

the GCN model to account for different types of relations in the graph. Fig. 7 illustrates the influence distributions for the multi-relational Yelp and Amazon datasets. It is evident that accounting for multiple relationships results in a rightward (positive) shift in the influence distributions. Despite the relational capabilities of R-GCN, our model demonstrates superior performance on both datasets.

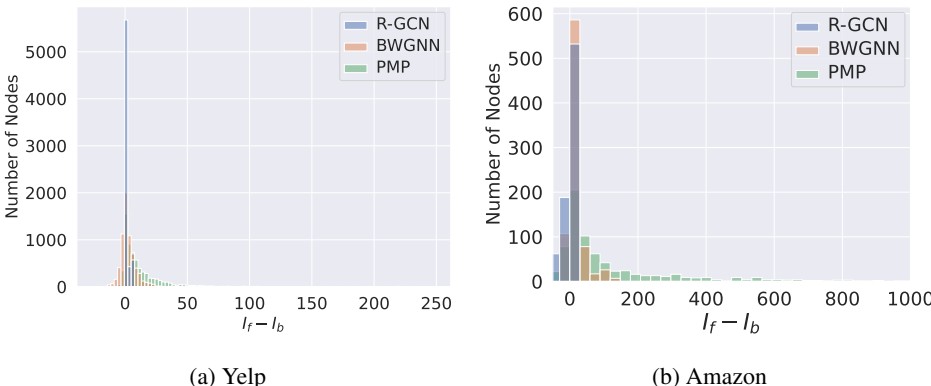

(a) Yelp                    (b) Amazon

Figure 7: Influence distribution.

