# OpenReview forum: "Partitioning Message Passing for Graph Fraud Detection"
_ICLR.cc/2024/Conference — ICLR 2024 poster_

### Official Review · Reviewer_wN2Y · 2023-10-23

**Soundness:** 2 fair
**Presentation:** 2 fair
**Contribution:** 2 fair
**Rating:** 5
**Confidence:** 4

**Summary:**

This paper discusses the challenges in applying Graph Neural Networks (GNNs) to Graph Fraud Detection (GFD) tasks, specifically the issues of label imbalance and the mixture of homophily and heterophily. While existing GNN-based GFD models typically exclude heterophilic neighbors during message passing, the authors argue for distinguishing neighbors with different labels instead of exclusion. They introduce a new approach called Partitioning Message Passing (PMP), which adapts the information aggregated from heterophilic and homophilic neighbors, preventing the model gradient from being dominated by benign nodes. Theoretical connections and extensive experiments demonstrate that PMP significantly enhances GFD task performance, effectively addressing these challenges.

**Strengths:**

1. This paper maintains a high level of self-containment and coherence, as the authors support the claims made within the manuscript through detailed explanations and experimental validation.
2. This paper addresses the issues of label imbalance and the mixture of homophily-heterophily by introducing a new approach that assigns distinct parameter matrices to neighbors from different classes.
3. The comparison algorithms used in experiments are state-of-the-art.

**Weaknesses:**

1. There are some reservations regarding the novelty of this paper for the following reasons:

     a) The issues of label imbalance and homophily-heterophily have been effectively addressed in the past. Notably, Tang et al. [1] offered a theoretical explanation for these concerns using spectral graph analysis.

     b) Building upon the findings of [1], this paper makes incremental enhancements in the field of fraud detection while questioning the complexity of earlier algorithms. Nonetheless, this paper does not provide comprehensive comparison of time and space complexities with previous works.

2. The approach of treating fraud detection as a binary classification problem has inherent limitations. Real-world scenarios often involve anomalies that cannot be neatly classified into a single class and that do not adhere to clustering assumptions. Given the multitude of existing papers that have already presented effective solutions to address the challenges of imbalance and homophily-heterophily by binary classification algorithms, it raises questions about the need for publishing similar articles at the cutting-edge conference ICLR.

3. An analysis about the labeled neighborhoods’ label distributions of central nodes in the training dataset is necessary. This is crucial as the proposed method relies heavily on the labels of neighboring nodes. Furthermore, it is important to emphasize the ratio of normal nodes to anomaly nodes in the training data.

4. Concerns have been raised about the correctness of equation (7).

[1] Tang, Jianheng, et al. Rethinking graph neural networks for anomaly detection. ICML, 2022.

**Questions:**

Please refer to the weakness.

---

> ### Author Response · Authors · 2023-11-18
> **Response to Reviewer wN2Y (part 1/3)**
>
> We thank the reviewer for providing insightful and constructive comments. We address the reviewer's comments/concerns as follows.
>
> **Q1.1: The issues of label imbalance and homophily-heterophily have been effectively addressed in the past. Notably, Tang et al. [1] offered a theoretical explanation for these concerns using spectral graph analysis.**
>
> **A1.1:** In our opinion, BWGNN proposed by Tang et al. [1] aims to preserve or emphasize information from certain frequencies based on the PDF of Beta distribution, so BWGNN acts as a band-pass spectral graph filter, which has been proved effective for graph with heterophily. However, BWGNN does not specifically cater to the label imbalance challenge. There are two key reasons for this. Firstly, BWGNN use Beta wavelets as propagation matrix, which is predefined and only related to the graph Laplacian, thus the propagation matrix of BWGNN is label-agnostic. Secondly, the feature transformation matrices of BWGNN is uniformly  shared across all nodes, so it is also label-agnostic. These two factors indicate that the training process of BWGNN does not  explicitly account for the label imbalance inherent in certain graphs. The empirical evidence presented in Table 4 of Section 5.3 illustrates that our approach more effectively mitigates problems arising from label imbalance compared to BWGNN. This is achieved by  investigating the influence distribution from neighbors on minority  class nodes and tailoring the message passing process accordingly.
>
> **Q1.2：Building upon the findings of [1], this paper makes incremental  enhancements in the field of fraud detection while questioning the  complexity of earlier algorithms. Nonetheless, This paper does not provide comprehensive comparison of time and space complexities with  previous works.**
>
> **A1.2:** We first show a detailed comparison of the time and space complexities in the following table (PC-GNN is built upon Care-GNN, and GHRN is an extension of BWGNN. They share similar complexity profiles with their respective base models), where $|E| \ll N^2$ is the number of edges, $d$ is the feature dimension and $C$ is the number of spectral filter in BWGNN. Usually, $N > |E|$, thus our model demonstrates optimal time and space complexities compared to these baselines. We have updated it in the Appendix A of the revised manuscript.
>
> |              | Time Complexity | Storage Cost        |
> | ------------ | --------------- | ------------------- |
> | Care-GNN     | $O(Nd + \|E\|)$   | $O(Nd + N^2 + \|E\|)$ |
> | H2-FDetector | $O(N^2)$        | $O(Nd + N^2)$       |
> | BWGNN        | $O(C\|E\|)$       | $O(Nd + C\|E\|)$      |
> | PMP (Ours)   | $O(\|E\| + N)$    | $O(Nd + \|E\|)$       |
>
> Besides, we would like to explain that our PMP is not built upon the findings of BWGNN. PMP and BWGNN have different motivation, methodology, and theoretical insights. So we think BWGNN cannot hide our contributions. The main differences between PMP and BWGNN are summarized as follows:
>
> **[Motivation]** PMP is motivated by investigating the influence propagation during message passing (Section 2.2 in the updated version), which leads us to propose a fundamental principle for the  successful application of GNNs in GFD: distinguishing between neighbors with different labels during the message passing. However, BWGNN is motivated by the 'right-shift' phenomenon of the spectral energy distribution, which inspires the BWGNN to define a band-pass graph filter based on beta wavelet.
>
> **[Methodology]** （1）**Label Utilization**: PMP is spatial and label-aware,  using distinct weight matrices for different labels. BWGNN, a spectral  model, has shared, label-agnostic parameters.(2). **Weight Matrix Design**: PMP's weight matrices are root-specific (See Section 3), tailored for each node. BWGNN uses randomly initialized weights. (3) **Unlabeled Nodes Handling**: PMP treats unlabeled nodes as a mix of benign and fraud labels (See Section 3),  while BWGNN treats all nodes uniformly, regardless of their labeled or unlabeled status.

---

> ### Author Response · Authors · 2023-11-18
> **Response to Reviewer wN2Y (part 2/3)**
>
> **Q2: The approach of treating fraud detection as a binary classification problem has inherent limitations. Real-world scenarios often involve anomalies that cannot be neatly classified into a single class and that do not adhere to clustering assumptions. Given the multitude of existing papers that have already presented effective solutions to address the challenges of imbalance and homophily-heterophily by binary classification algorithms, it raises questions about the need for publishing similar articles at the cutting-edge conference ICLR.**
>
> **A2:** We acknowledge that the binary classification framework has its limitations in capturing the full spectrum of real-world frauds. However, in real-world application, due to the large scale and strong heterogeneity of the network, we usually need to extract all potential fraud nodes first, and then do a more detailed classification and analysis of the extracted fraud nodes. Thus, modeling GFD problem as a binary classification is usually the first step. Besides, to our best knowledge, all the datasets publicly available and commonly used in research, including those we have utilized, are modeled for binary classification. This modeling choice reflects the practical realities of many real-world applications, where a binary approach remains a fundamental and highly effective method for initial fraud detection. We still greatly appreciate your valuable suggestion. Considering the current limitation of most works focusing on binary classification, we plan to explore the issue of multi-classification in our subsequent work.
>
> Given the practical importance of graph fraud detection task, we agree that this problem has been studied in many past work. However, we do not think research in this area have converged because this problem is highly relevant to industrial applications. This requires that the model can capture the essence of the problem and be simple enough to make it easy to deploy in industrial environment. In our paper, we try to explore the key to solving this problem, rather than blindly designing complicated algorithms. Based on the motivation study, we propose that the key to successfully applying GNNs for GFD is to distinguish neighbors with different labels during message passing, instead of excluding heterophilic neighbors used in most prior work. Our proposed PMP is built upon this finding, and maintains simplicity to facilitate ease of deployment in industrial applications. To verify this point,  in Appendix E.2 of our paper, we present experiments conducted on datasets provided by our industrial collaborators. These experiments demonstrate the applicability and effectiveness of our method in real-world industrial scenarios, affirming its practical relevance. Thus, we believe our paper offers valuable insights for future research in  this domain.

---

> ### Author Response · Authors · 2023-11-18
> **Response to Reviewer wN2Y (part 3/3)**
>
> **Q3: An analysis about the labeled neighborhoods’ label distributions of  central nodes in the training dataset is necessary. This is crucial as  the proposed method relies heavily on the labels of neighboring nodes.  Furthermore, it is important to emphasize the ratio of normal nodes to anomaly nodes in the training data.**
>
> **A3:** Your suggestion is indeed crucial.  Therefore, we present the label distributions of the labeled neighborhoods of the training nodes in the **Appendix D.2** of the revised paper. Here, we present these distributions in it in tabular form. More details and distribution diagram can be found in the Appendix D.2.
>
> To analyze the label distributions of the labeled neighborhoods of training nodes, we use $\frac{|\mathcal{N}\_{fr}|}{|\mathcal{N}\_{be}|}$ to represent the number of fraud neighbors relative to benign neighbors for a central node. $\frac{|\mathcal{N}\_{fr}|}{|\mathcal{N}\_{be}|} <1$  indicates that the number of labeled fraud neighbors is fewer than benign neighbors. Specifically, if $\frac{|\mathcal{N}\_{fr}|}{|\mathcal{N}\_{be}|} <0.5$, it  indicates that the central node has far fewer fraud neighbors compared  to benign ones, denoting a highly imbalanced neighborhood.  Consequently, the smaller the value of $\frac{|\mathcal{N}\_{fr}|}{|\mathcal{N}\_{be}|}$, the more imbalanced the label distribution of neighbors of a given central node is. We use $|V_{\frac{|\mathcal{N}\_{fr}|}{|\mathcal{N}\_{be}|} < 1}|$ to represent the number of training center nodes with $\frac{|\mathcal{N}\_{fr}|}{|\mathcal{N}\_{be}|} < 1$, allowing us to assess the label distribution relative to this ratio.
>
> |                                                              | Yelp  | Amazon | T-Finance |
> | ------------------------------------------------------------ | ----- | ------ | --------- |
> | $\frac{\|V_{\frac{\|\mathcal{N}\_{fr}\|}{\|\mathcal{N}\_{be}\|} < 1}\|}{N}$ | 98.7% | 99.9%  | 100%      |
> | $\frac{\|V_{\frac{\|\mathcal{N}\_{fr}\|}{\|\mathcal{N}\_{be}\|} < 0.5}\|}{N}$ | 97.2% | 99.7%  | 100%      |
>
> Our results show that commonly used datasets such as Yelp, Amazon, and T-Finance suffer from significant label imbalance in the neighborhoods, which could lead to the general vanishing of the general GNN model with respect to the minority class neighbors, as we state in Section 2.2. Thus, since the model trained under the condition of imbalanced neighborhood label distribution, the ideal model needs to enhance the influence from the minority class neighbors on the center nodes. As demonstrated in Section 5.3, our PMP method effectively achieves  this, enhancing the influence from minority class neighbors more  efficiently than traditional GNN approaches.
>
> Regarding your query about the ratio of normal to anomaly nodes in the training data, this ratio in the training set is the same as the ratio in the whole graph. Specifically, in the following table, we show the ratio of fraud nodes in the whole graph. For example, 14.53% of nodes in the Yelp are fraud nodes, so in the training set, the fraud nodes also occupy 14.53% of the training set.
>
> |           | % Fraud Nodes |
> | --------- | ------------- |
> | Yelp      | 14.53%        |
> | Amazon    | 6.87%         |
> | T-Finance | 4.58%         |
> | T-Social  | 3.01%         |
>
> **Q4: Concerns have been raised about the correctness of equation (7).**
>
> **A4:** Based on your concerns, we re-examined the derivation process of Theorem 1 and equation (7). The proof of equation (7) is similar to the spectral analysis flow of traditional GCN, and the detailed proof can be found in Appendix B. We're sure it's correct.

---

> > ### Comment · Reviewer_wN2Y · 2023-11-20
> > **Thanks for the response**
> >
> > Thank you for the author's response. While your reply has addressed some of my concerns, I still find this article slightly below the borderline accept threshold. The reasons are outlined below:
> >
> > 1. The main concept of creating a spatial graph convolution as a high- and low-pass filter is intriguing, but the methodology appears somewhat rough. For instance, relying on the output of the $f_{self}$ function as input for an MLP to learn $\alpha^{l}_i$ lacks convincing rationale, particularly since the authors did not analyze the characteristics of node features for normal and abnormal nodes.
> >
> > 2. In the industry, treating fraud detection as a binary classification problem may lead to model failure, as fraud patterns are not static. The binary classification model can become invalid when the characteristics of new fraudsters differ significantly from those of old fraudsters.
> >
> > 3. Equation 12 in Algorithm A reveals that the model still grapples with imbalance issues. While the author addresses the problem of unbalanced label distribution among neighbouring nodes through PARTITIONING MESSAGE PASSING, the imbalance of the central node persists. While PARTITIONING MESSAGE PASSING may alleviate the negative impact of label imbalance to some extent, it does not provide a direct solution.
> >
> > 4. In real-world scenarios, it is possible that many labelled central nodes may lack labelled fraudulent neighbours, due to the scarcity of fraud nodes. I am uncertain if this situation affects the model's performance. I previously suggested that the author analyse the labelled neighbourhood of the central node to address this concern.
> >
> > To sum up, I will keep the score 5.

---

> > > ### Author Response · Authors · 2023-11-22
> > > **Response to follow-up questions (part 1/2)**
> > >
> > > Dear Reviewer wN2Y,
> > >
> > > Thank you very much for your reply and follow-up questions. We address your questions as follows:
> > >
> > > **Q1: The main concept of creating a spatial graph convolution as a high- and  low-pass filter is intriguing, but the methodology appears somewhat rough. For instance, relying on the output of the $f_{self}$ function as input for an MLP to learn $\alpha_i^l$ lacks convincing rationale, particularly since the authors did not analyze the characteristics of node features for normal and abnormal nodes.**
> > >
> > > **A1:**  To clarify, in the $(l+1)$-th layer,  the weight matrix $\mathbf{W}\_{\text{un}}^{(l)} = \alpha^{(l)}\_i \mathbf{W}\_{\text{fr}}^{(l)} + (1 - \alpha^{(l)}\_i) \mathbf{W}\_{\text{be}}^{(l)}$ is employed to aggregate unlabeled neighbors of $\mathbf{h}\_i^{(l)}$ for generating the node representation $\mathbf{h}\_i^{(l+1)}$. $\alpha^{(l)}\_i = \Phi(\mathbf{h}^{(l)}_i)$ , as a function of the output representation of the previous layer. It is reasonable since nodes are typically categorized as either fraud or benign, this approach is deliberately designed to avoid treating unlabeled neighbors as a distinct, new class. Instead, it allows the model to interpret these unlabeled neighbors as a blend of the known classes (fraud and benign), reflecting the real-world ambiguity in their classification. Furthermore, setting $\alpha^{(l)}_i$ as a function of the center node is a strategic decision, because different center nodes should emphasize different kind of information from its neighborhoods. Thus the strategy for handling neighbors should be contingent on the characteristics of the center node.
> > >
> > > Here we provide the analysis for the characteristics of fraud and benign nodes to show that $\alpha^{(l)}\_i = \Phi(\mathbf{h}^{(l)}\_i)$ is different between the center fraud nodes to the center benign nodes. Since the function $\Phi(\cdot)$ is applied on the hidden feature of each node, for clear, here we show the $l_2$ distance between the average representation of the hidden fraud nodes $\mathbf{h}^{(l)}\_{\mathbf{fr}} = \sum_{\\{v_i | y_i =1\\}}  \mathbf{h}^{(l)}\_{i}$ , and hidden benign nodes $\mathbf{h}^{(l)}\_{\mathbf{be}} = \sum_{\\{v_i | y_i =1\\}}  \mathbf{h}^{(l)}_{i}$, and their distance is $D^{(l)} = \|\|\mathbf{h}^{(l)}\_{\mathbf{fr}} - \mathbf{h}^{(l)}\_{\mathbf{be}}\|\|_2$.
> > >
> > > |           | Yelp   | Amazon   |
> > > | --------- | ------ | -------- |
> > > | $D^{(0)}$ | 0.3283 | 247.6826 |
> > > | $D^{(1)}$ | 5.7746 | 657.8542 |
> > >
> > > From the above table, we can find that no matter which layer of output it is, there is a big difference in the characteristics between fraud nodes and benign nodes. It indicates that $\alpha^{(l)}_i = \Phi(\mathbf{h}^{(l)}_i)$ can generate different values for the center nodes with different labels, thus the model handles the neighbors during message passing based on the center nodes. Besides, the difference between fraud nodes and benign nodes become greater from the $0$-th layer to the first layer, which shows that the distinguishability between nodes with different classes increase after PMP layer.
> > >
> > > **Q2: In the industry, treating fraud detection as a binary classification problem may lead to model failure, as fraud patterns are  not static. The binary classification model can become invalid when the characteristics of new fraudsters differ significantly from those of old fraudsters.**
> > >
> > > **A2:** We recognize and agree with your concern. However, our approach to model fraud detection as a binary classification task is primarily driven by the current landscape of available datasets and the state-of-the-art methodologies in this domain. As you rightly pointed out, the characteristics of new fraudsters can significantly differ from those of historical ones, making the development of a robust multi-class fraud detection model more relevant and necessary. Unfortunately, the scarcity of publicly available, multi-class fraud detection datasets has been a major limiting factor. Since prior work such as  BWGNN (ICML 2022), PC-GNN (WWW 2021), and GHRN (WWW 2023), which also model GFD as a binary classification  problem, categorizing nodes simply as 'fraudulent' (label 1) or 'benign' (label 0), so we think our work aligns with recent advancements and standard practices in the field.
> > >
> > > We very much agree with your perspective on the importance and relevance of multi-class fraud detection. We acknowledge that this approach represents a more realistic and meaningful scenario in fraud detection. Moving forward, we are committed to exploring multi-class fraud detection in our future work.

---

> > > ### Author Response · Authors · 2023-11-22
> > > **Response to follow-up questions (part 2/2)**
> > >
> > > **Q3: Equation 12 in Algorithm A reveals that the model still grapples with  imbalance issues. While the author addresses the problem of unbalanced  label distribution among neighbouring nodes through PARTITIONING MESSAGE PASSING, the imbalance of the central node persists. While PMP may alleviate the negative impact of label imbalance to some extent, it does not provide a direct solution.**
> > >
> > > **A3:**  In our approach, two parameter matrices are universally applied across all nodes: $f_{self}$ , which transforms the input feature into the embedding space, and an MLP, as detailed in Equation 12, which projects the node embedding from the embedding space to the label space. The imbalance problem of center nodes is usually alleviated by using weighted loss function, i.e.,  assigning greater weight to the training minority class nodes in the loss function, which is a common setting in imbalanced learning problem. In our code (https://anonymous.4open.science/r/PMP_submit-F35D/training_procedure/prepare.py), we also choose to use this setting as follows:```loss_func = nn.CrossEntropyLoss(weight = torch.tensor([1., weight]))```, where the weight is inversely proportional to the number of nodes.
> > >
> > > We do not discuss it in the paper, as it is a default setting in much of the imbalanced learning literature. We believe that the performance challenges GNNs face on imbalanced  graphs primarily stem from the imbalance in local neighbor aggregation  rather than the global label distribution imbalance. To illustrate this, we refer to Table 3 in our ablation study, where GraphSAGE, trained using an imbalanced loss function, achieves an AUC of 89.38%  on the Yelp dataset. By integrating our partitioning message-passing  training strategy, there is a notable performance improvement of 3.77%.  Here we remove the weight in the loss function from GraphSAGE and PMP to show the influence of weighted loss function on the performance.
> > >
> > > |                                                | Yelp  | Amazon |
> > > | ---------------------------------------------- | ----- | ------ |
> > > | GraphSAGE (without weighted_loss)              | 88.46 | 93.10  |
> > > | GraphSAGE + weighted_loss                      | 89.38 | 93.16  |
> > > | GraphSAGE  (without weighted_loss) + partition | 93.18 | 96.07  |
> > > | GraphSAGE + weighted_loss + partition          | 93.15 | 96.33  |
> > >
> > > From the table, it is observable that the models incorporating weighted  loss show only a marginal difference in performance compared to those without it. It shows that the primary challenge in GNN performance on imbalanced graphs lies in the local neighbor aggregation process.
> > >
> > > We apologize for the confusion and will clarify the usage of the weighted loss in the revised manuscript.
> > >
> > > **Q4: In real-world scenarios, it is possible that many labelled central nodes may lack labelled fraudulent neighbours, due to the scarcity of fraud nodes. I am uncertain if this situation affects the model's performance. I previously suggested that the author analyse the labelled neighbourhood of the central node to address this concern.**
> > >
> > > **A4:** We understand  your concern regarding the potential scarcity of fraudulent neighbors  for labeled central nodes, which could potentially impact the training of $\mathbf{W}\_{\text{fr}}$ . In our model, for nodes without labeled fraud neighbors, its benign neighbors are still processed through $\mathbf{W}\_{\text{be}}$. For its unlabeled neighbors, we assume these neighbors comprise a mix of fraud and benign nodes, thus these neighbors are transformed by $\mathbf{W}\_{\text{un}} = \alpha \mathbf{W}\_{\text{fr}} + (1 - \alpha) \mathbf{W}\_{\text{be}}$, so optimizing $\mathbf{W}\_{\text{un}}$ indirectly contributes to the training of $\mathbf{W}\_{\text{fr}}$. Meanwhile, for nodes that do have labeled fraud neighbors, $\mathbf{W}\_{\text{fr}}$ is directly optimized. The below table are the statistics showing the proportion of training nodes with labeled fraud neighbors across different datasets, where $N_{fr}$ is the number of training nodes that have at least 1 labeled fraud neighbors, and $N$ is the number of training nodes.
> > >
> > > |           | Proportion of Nodes with labeled Fraud Neighbors ($\frac{N_{fr}}{N}$) |
> > > | --------- | ------------------------------------------------------------ |
> > > | Yelp      | $\frac{18007}{18381}$                                        |
> > > | Amazon    | $\frac{3352}{3455}$                                          |
> > > | T-Finance | $\frac{11611}{15742}$                                        |
> > >
> > > This statistics (under 40% training ratio) indicate indicate that a majority of the training nodes have labeled fraud neighbors, allowing for direct optimization of $\mathbf{W}\_{\text{fr}}$ in many cases. Therefore, in the training phase, we ensure that both $\mathbf{W}\_{\text{fr}}$ and $\mathbf{W}\_{\text{un}}$ are adequately trained, mitigating the concern about the scarcity of fraud neighbors and its potential impact on the model's performance.

---

### Official Review · Reviewer_XqA8 · 2023-10-31

**Soundness:** 2 fair
**Presentation:** 2 fair
**Contribution:** 2 fair
**Rating:** 5
**Confidence:** 4

**Summary:**

This work addresses the challenges of label imbalance and the complex interplay between homophily and heterophily in Graph Neural Networks for Graph Fraud Detection. It introduces a novel approach called Partitioning Message Passing (PMP). In this method, neighboring nodes of different classes are processed using distinct, node-specific aggregation functions. Furthermore, the central node has the ability to adaptively fine-tune the information it gathers from both its heterophilic and homophilic neighbors. Empirical results reveal that the PMP method outperforms other competitive algorithms across a range of datasets, including Yelp, Amazon, and T-Finance, while maintaining an optimal balance between performance and computational time.

**Strengths:**

1. The PMP method is well-designed, offering a straightforward solution that is easy to understand and implement.
2. The article employs a comprehensive analytical framework to validate the effectiveness of the PMP method, adding to its credibility.

**Weaknesses:**

1. The primary innovation in the PMP method lies in the use of different weighting matrices for aggregating nodes of various classes. While effective, this focus may be perceived as lacking in breadth in terms of overall innovativeness.
2. The article would benefit from a more meticulous attention to the use of symbols and language. Ensuring consistent and clear terminology and notation would contribute to the paper's readability and accessibility.

**Questions:**

1. On Page 2, in the penultimate line of "where each node v_i is assigned a binary label y_i∈Y", please confirm whether it is "Y" or "fi"?
2. In the description of Eq.(4) on page 4, it is mentioned that "In other words, a small α_i^((l)) means that the model treats unlabeled neighbors more similarly to fraud nodes". In conjunction with Eq.(4), shouldn't it be the case that a smaller α_i^((l)) makes unlabeled neighbors more biased to benign nodes?
3. Throughout the paper, sections 2, 3, and 7 are more similar to one part of the content, related work, could they be synthesized into one section?
4. When introducing PMP, the article mentions the use of one layer of MLP in the generation of α_i^((l)). Still, it does not note how many layers of MLP are used in the subsequent "Root-specific weight matrices generation.
5. In the experimental results for the T-Social dataset, the AUC value is much higher than that of the comparative algorithm, close to 100%, is it possible to analyze the reason for this excellent result?
6. In the last sentence of Section 6.4, "wherein each of these designs brings more than a 1% improvement across most metrics", does that mean that each approach delivers at least a 1% improvement over GraphSAGE, or does it mean that it provides a 1% improvement over the previous pivotal components? Please be as precise as possible.

---

> ### Author Response · Authors · 2023-11-18
> **Response to Reviewer XqA8 (part 1/2)**
>
> We thank you for your time and effort in reviewing our paper. Below, we respond to address your concerns.
>
> **Q1: The primary innovation in the PMP method lies in the use of different weighting matrices for aggregating nodes of various classes. While effective, this focus may be perceived as lacking in breadth in terms of overall innovativeness.**
>
> **A1:** We understand that our embarrassingly simple model may give rise to concerns about its innovativeness. We respectfully argue that our work is novel, and we highlight our technical novelty as follows:
>
> 1. **[High-level insight]** We propose that the key to successfully applying GNNs for GFD relies not on excluding which used in most prior work, but on distinctly recognizing and processing neighbors with different labels during message passing.  Thus, our approach is a fundamental departure from existing methodologies.
>
> 2. **[Strong practical implications in applying GNN to GFD Applications]**  Graph fraud detection is a practical problem, especially in the industrial application. Thus, it is crucial to capture the essence of the problem, and design a simple and easy-to-deploy model. Our PMP method is designed with these requirements in mind. Through the motivation study, theoretical analysis and strong empirical results, we demonstrate PMP's effectiveness. The simplicity of our model does not diminish its impact; rather, it enhances its practical utility, making it a valuable tool in real-world applications.
>
> While PMP's design is straightforward in form, it represents a meaningful step forward in the field, offering both a novel perspective and a practical solution. We believe that PMP has a certain guiding role in future work.
>
> **Q2: The article would benefit from a more meticulous attention to the use of symbols and language. Ensuring consistent and clear terminology and  notation would contribute to the paper's readability and accessibility.**
>
> **A2:** Thanks for your valuable feedback. We have undertaken a thorough review of our manuscript with a focus on standardizing the terminology and symbols used throughout the paper. The modifications have been highlighted in blue in the resvised manuscript.
>
> **Q3: On Page 2, in the penultimate line of "where each node v_i is assigned a binary label $y_i\in Y$", please confirm whether it is "Y" or "fi"?**
>
> **A3:** In our paper, $\mathcal{Y} = \{y_1, y_2, \cdots, y_N\}$ represents the set of node labels, where $y_i \in \mathcal{Y}$ means the label of node $v_i$. $y_i$ has two possible values, i.e., $y_i = 1$ denotes $v_i$ is a fraud node $0$ denotes benign node.
>
> **Q4:  In the description of Eq.(4) on page 4, it is mentioned that "In other words, a small $\alpha_i^{(l)}$ means that the model treats unlabeled neighbors more similarly to fraud nodes". In conjunction with Eq.(4), shouldn't it be the case that a smaller $\alpha_i^{(l)}$ makes unlabeled neighbors more biased to benign nodes?**
>
> **A4:** Thanks for pointing this out. Upon re-examination, we agree that there was an error in our description of how $\alpha^{(l)}\_{i}$ influences the treatment of unlabeled neighbors in our model. As you correctly noted, a small $\alpha^{(l)}\_{i}$ should indeed indicate that the model treats unlabeled neighbors more similarly to benign nodes due to the larger weight placed on $\mathbf{W}\_{\text{be}}^{(l)}$. We have corrected the text in the manuscript to accurately reflect this relationship.
>
> **Q5: Throughout the paper, sections 2, 3, and 7 are more similar to one part of the content, related work, could they be synthesized into one section?**
>
> **A5:** We agree your suggestion that enhances the readability. We have merged Sections 2, 3, and 7 into a single cohesive section titled "Background and Motivation" in new Section 2.
>
> **Q6:  When introducing PMP, the article mentions the use of one layer of MLP in the generation of $\alpha^{(l)}_{i}$. Still, it does not note how many layers of MLP are used in the subsequent "Root-specific weight matrices generation".**
>
> **A6:** To maintain simplicity in the model design, we also apply the single-layer MLP in the "Root-specific weight matrices generation" to define the weight generators $\Psi_{\text{fr}}$ and $\Psi_{\text{be}}$. We have clarified this point in the revised manuscript. Moreover, we show influence of the number of MLP layers on the final results (AUC). We can find that the number of MLP layers does not drastically affect the performance of the model.
>
> | #layers | Yelp  | Amazon |
> | ------- | ----- | ------ |
> | 1       | 93.97 | 97.57  |
> | 2       | 93.95 | 97.51  |

---

> ### Author Response · Authors · 2023-11-18
> **Response to Reviewer XqA8 (part 2/2)**
>
> **Q7:  In the experimental results for the T-Social dataset, the AUC value is much higher than that of the comparative algorithm, close to 100\%, is it possible to analyze the reason for this excellent result?**
>
> **A7:** Referring to Table 2, our PMP model shows substantial improvements compared to other baselines. Specifically, it surpasses the current state-of-the-art by 4.9% in AUC and achieves an 11.21% relative increase in Macro-F1 score.
>
> We analyze the characteristics of the T-Social dataset. From the global perspective, as shown in Table 4, T-Social is an exceptionally large-scale dataset, having over 5 million nodes.This size is roughly 100 times greater than other datasets used in our study, presenting a unique scalability challenge for GNNs. Some baselines (Care-GNN, PC-GNN, H2-FDetector, GHRN) focus on augmenting graph structure by resampling neighbors, reweighting or pruning edges for each node to balance the message passing. However, given the immense size of T-Social, coupled with its diverse local patterns (where node degrees range from 1 to 3217), it becomes challenging to learn or define a general and shared strategy for local structure augmentation applicable to all nodes. Consequently, these models are less effective on large scale graphs. Differently, PMP and BWGNN do not make any assumptions about the local structure at the node level. They learn directly from the original graph without implementing any node-level augmentations. These approaches effectively alleviates the scalability challenges posed by the size and complexity of T-Social.
>
> Then we would like to analyze the reason why PMP is better than BWGNN from the perspective of label imbalance. As shown in Table 4, T-Social exhibits a higher imbalance level, with only 3% of its nodes labeled as fraud. This leads to benign nodes overwhelmingly dominate the dataset, both in number and proportion. For BWGNN, the propagation matrix is predefined by Beta wavelet matrix, which is label-agnostic, and the feature transformation matrix is shared across all nodes, it leads to label imbalance knowledge can not be encoded into the model parameters during message passing. For our PMP, our model encode label information into the trainable parameter by utilizing distinct transformation matrices for different classes during the aggregation process, By processing nodes of different classes independently, the overfitting of the model to the class is alleviated. We believe that this is the key factor contributing to the excellent results achieved by our PMP.
>
> **Q8: In the last sentence of Section 5.4, "wherein each of these designs brings more than a 1% improvement across most metrics", does that mean that each approach delivers at least a 1% improvement over GraphSAGE, or does it mean that it provides a 1% improvement over the previous pivotal components? Please be as precise as possible.**
>
> **A8:** Thanks for pointing out our imprecise description. The "+partition" serves as the foundational element and is responsible for the major improvements observed, because it yields more than 3% improvement compared to GraphSAGE on most metrics. The subsequent components, "++adaptive combination" and "+++root-specific weights", are developed upon the "+partition" base. 1% improvement here means "++adaptive combination" and "+++root-specific weights" bring a cumulative improvement of more than 1% to "+partition". Both  components provide useful contributions, but the amount of contribution provided by the two components differs depending on the datasets. For example, "++adaptive combination" bring a small improvement while "+++root-specific weights" provide more, such as in T-Finance dataset, while in Amazon dataset, "++adaptive combination" brings more improvement. We recognize that our initial expression in the paper might have been  imprecise, and we have now amended as:
>
> **The "++adaptive combination", building upon the "+partition", brings incremental improvements across all datasets. The "+++root-specific weights", further extending the preceding component, enhances performance on most metrics in most datasets. Combined, these two components contribute to a cumulative improvement of more than 1% over the "+partition" baseline.**
>
> The modification has been marked as blue in Section 5.4 in the revised manuscript.

---

> ### Author Response · Authors · 2023-11-22
> **Looking forward to your response**
>
> Dear Reviewer XqA8,
>
> We appreciate your valuable feedback on our paper. With the discussion period nearing its end, we've compiled a succinct summary of our responses to your comments for ease of reference in your final evaluation:
>
> **Part 1**: We highlight our contribution from the perspectives of high-level insight and practical value.
>
> **Part 2**: The manuscript have been revised to ensure consistency and enhance its readability and accessibility.
>
> **Part 3**: We have corrected an inaccurate description of $\alpha$.
>
> **Part 4**: The sections of the paper have been reorganized based on your suggestions.
>
> **Part 5**: We have added detailed hyperparameter settings.
>
> **Part 6**: We analyzed the experimental results on T-Social datasets, and modified  imprecise descriptions in the experiment analysis.
>
> Warm regards,
>
> The Authors of Submission 9114.

---

> > ### Comment · Reviewer_8onS · 2023-11-23
> > **Thanks for the response**
> >
> > Thanks for the author's response especially the detailed discussion on the differences between ACM and the proposed PMP. These answers addressed my concerns, so I would like to increase my rating.

---

> > ### Comment · Reviewer_XqA8 · 2023-11-23
> >
> > Dear authors,
> > Thank you for your response and pardon my late reply. I appreciate the time and efforts you put on rebuttal. The response is clear and well addressed my concerns. Therefore, I am amending my score upward.

---

### Official Review · Reviewer_8onS · 2023-11-01

**Soundness:** 2 fair
**Presentation:** 3 good
**Contribution:** 2 fair
**Rating:** 6
**Confidence:** 4

**Summary:**

In this paper, the authors address fundamental challenges faced in applying Graph Neural Networks (GNNs) to Graph Fraud Detection (GFD) tasks, namely, label imbalance and the mixture of homophily-heterophily. Existing GNN-based GFD models modify graph structures to accommodate GNNs' homophilic bias by excluding heterophilic neighbors during message passing. However, the authors propose a novel perspective: instead of excluding, they advocate for distinguishing neighbors with different labels. They introduce a method called Partitioning Message Passing (PMP), a message passing paradigm tailored for GFD. In PMP, neighbors with different classes are aggregated using distinct node-specific aggregation functions. This approach allows the central node to adaptively adjust the information gathered from both heterophilic and homophilic neighbors. By doing so, PMP prevents the model gradient from being dominated by benign nodes, which constitute the majority of the population. The authors establish a theoretical connection between the spatial formulation of PMP and spectral analysis, characterizing PMP as an adaptive node-specific spectral graph filter. This demonstrates PMP's ability to handle graphs with mixed heterophily and homophily. Extensive experiments validate the effectiveness of PMP, showing significant performance improvements in Graph Fraud Detection tasks. PMP's innovative approach of distinguishing rather than excluding neighbors with different labels showcases its potential in enhancing the capabilities of GNNs for fraud detection on graphs.

**Strengths:**

- The solution to adaptively learn from heterophilous and homophilous nodes for fraud detection is interesting and the theoretical analysis is sound.

- The paper is generally well-written and almost clear everywhere.

- Experiments conducted on datasets with different sizes show the effectiveness and efficiency of the proposed method in graph fraud detection.

**Weaknesses:**

- The relationships between heterophily and imbalance (which is specific in fraud detection) are not clear. This is important to understand the problem.

- The relationships between the proposed method and some previous spectral GNNs, e.g., [1], have not been discussed. A lack of discussions about differences may limit the novelty of the proposed method.

- A minor issue: repeated references 2nd and 3rd articles.

[1] Is Heterophily A Real Nightmare For Graph Neural Networks on Performing Node Classification? 2021

**Questions:**

- The strategy of partitioning message passing is very similar to GNNs with adaptive channel mixing used in [1] although [1] is from the spectral perspective. It will be interesting to discuss the differences and relationships between your proposed method and [1].

- The discussion on the relationships between heterophily and imbalance is not detailed. A detailed empirical and/or theoretical analysis of relationships between heterophily and imbalance on graphs should be conducted to better understand the problem.

[1] Is Heterophily A Real Nightmare For Graph Neural Networks on Performing Node Classification? 2021

---

> ### Author Response · Authors · 2023-11-18
> **Response to Reviewer 8onS (Part 1/2)**
>
> We thank you for your time and effort in reviewing our paper. Below, we respond to address your concerns.
>
> **Q1: The relationships between heterophily and imbalance (which is specific in fraud detection) are not clear. This is important to understand the problem.**
>
> **A1:** In the context of network fraud detection, attackers strategically inject fraud nodes sparsely within benign communities to camouflage fraudulent activities and spread their influence to normal nodes. As a result, these fraud nodes frequently establish connections with benign nodes, leading to a network characterized by heterophily -- where connections are predominantly between nodes with dissimilar labels (i.e., fraud nodes connecting with benign nodes). In the **Table 5 of Appendix D.1**, we show the heterophily level with measure $\hat{\eta}$ of real-world datasets , which clearly demonstrate the heterophilic nature of such networks.
>
> Furthermore, the injection of fraud nodes into the network is deliberately kept to a minimum to conceal their fraudulent activities. This results in a significantly smaller population of fraud nodes compared to benign ones, thus engendering a label imbalance in the fraud graph. As evidenced in Table 5 of Appendix D.1, less than 10% of nodes in commonly used datasets are labeled as fraud nodes. Here we show the statistical data of the benchmark datasets as follows:
>
> |           | Imbalance Level (% Fraud Nodes) | Homophily ratio ($\hat{\eta}$) |
> | --------- | ------------------------------- | ------------------------------ |
> | Yelp      | 14.53%                          | 0.0538                         |
> | Amazon    | 6.87%                           | 0.0512                         |
> | T-Finance | 4.58%                           | 0.4363                         |
> | T-Social  | 3.01%                           | 0.1003                         |
>
> Overall, such a sparse distribution of fraud nodes not only confirms the label imbalance but also contributes to the network's heterophilic characteristics. Such interplay of heterophily and label imbalance in fraud graphs provides a unique challenge for GNNs.

---

> ### Author Response · Authors · 2023-11-18
> **Response to Reviewer 8onS (Part 2/2)**
>
> **Q2: The strategy of partitioning message passing is very similar to GNNs with adaptive channel mixing used in [1] although [1] is from the spectral perspective. It will be interesting to discuss the differences and relationships between your proposed method and [1].**
>
> **A2:** Thanks for your suggestions, which inspire us to rethink our Partitioning Message Passing (PMP) and Adaptive Channel Mixing (ACM) [1]. We agree that our work does indeed share similarities with ACM. However, the motivation, methodology, theoretical insights still differ. We provide a detailed comparison from these perspectives as follows:
>
> **[Motivation]** In our paper, we contend that the key to successfully applying GNNs for GFD task lies not in the exclusion of neighbors but in the ability to *distinguish* between them based on their labels during the neighborhood aggregation. This core belief is operationalized in our proposed PMP framework, which integrates this motivation into the traditional message passing paradigm. By partitioning neighbors during the aggregation phase using distinct transformation matrices, PMP effectively discriminates between different classes of neighbors, ensuring that the message passing mechanism is both simple and highly effective in addressing the unique challenges of GFD.
>
> Contrastingly, ACM is motivated by the concept of "*Diversification Distinguishability*", which primarily involves attributing negative weights to heterophilic neighbors through a spectral synthesis of low-pass and high-pass filters. This technique fundamentally differs from our PMP. It is because the use of negative weights in ACM is indicative of removing (or subtracting) the heterophilc components from the homophilic neighbors. This can be interpreted as a form of information '*forgetting*' where heterophilic features are de-emphasized in favor of homophilic features. It is different from PMP which preserves and utilizes all available information.
>
> **[Methodology]** ACM is rooted in spectral graph theory, while PMP is spatially oriented. This difference is not just theoretical but has practical implications. The spectral nature of ACM inherently influences its scalability, particularly in the context of large graphs. Typically, spectral-based models, including ACM, require the entire graph as input, which can pose challenges for mini-batch training and thereby limit scalability. In contrast, our PMP model, with its spatial-based framework, inherently supports mini-batch training and is thus more adaptable to large-scale graphs. Furthermore, considering the application of ACM in GFD tasks, its formulation, with removed nonlinearity for simplicity's sake, can be expressed as $H_{\textbf{ACM}}^{(l+1)} = \alpha F_{L}\begin{bmatrix}H^{(l)}\_{fr} \\\ H^{(l)}\_{be} \\\ H^{(l)}\_{un}\end{bmatrix} W_{1}^{(l)} + \beta F_{H}\begin{bmatrix}H^{(l)}\_{fr} \\\ H^{(l)}\_{be} \\\ H^{(l)}\_{un}\end{bmatrix} W_{2}^{(l)}$ where $F_L$ and $F_H$ are distinct spectral filters. We can observation that the trainable weight matrix $W_{L}^{(l)}$ and $W_{H}^{(l)}$ are both shared across all nodes. In contrast, our PMP model, when represented in matrix form, can be rewritten as $H_{\textbf{PMP}}^{(l+1)} = F \begin{bmatrix}H^{(l)}\_{fr} W_1\\\ H^{(l)}\_{be} W_2 \\\ H^{(l)}\_{un} (\alpha W_1 + (1-\alpha)W_2)\end{bmatrix}$ where $F$ is the normalized adjacency matrix. Unlike ACM, trainable weight matrices in PMP, $W_1$ and $W_2$ , are specifically applied to nodes with different labels, where $W_1$ and $W_2$ captures the information of fraud nodes and benign nodes respectively, reflecting a more label-aware and adaptive approach to message passing in the context of GFD.
>
> **[Theoretical Insights]** The theoretical insights of PMP and ACM reveal a significant divergence in their spectral capabilities, highlighting a fundamental difference in how they process graph signals. Specifically, ACM predominantly operates as a combination of low-pass and high-pass filters, which (de)emphasizing high or low frequency signal components can not control the information from other spectrum, i.e., ACM does not equate to the functionality of a band-pass filter. Differently, our theoretical analysis and visualization in Section.4 proves that our PMP operates as an adaptive graph filter which cover all frequencies in the spectral domain.
>
> We also think such comparisons are very meaningful for a deeper understanding of our work, thus we include them in **Appendix C** of the revised manuscript.
>
> **Q3: A minor issue: repeated references 2nd and 3rd articles.**
>
> **A3:** Thank you for pointing out the repeated references. We have carefully reviewed the manuscript and rectified the referencing. This correction has been reflected in the revised manuscript.

---

> ### Author Response · Authors · 2023-11-22
> **Looking forward to your response**
>
> Dear Reviewer 8onS,
>
> Thank you again for your thoughtful feedback on our submission. As the discussion period draws to a close, we have prepared a brief summary  below that encapsulates our responses to your points for your convenience in the final review:
>
> **Part 1**: We have explained the relationship between homophily and label imbalance under the fraud detection task.
>
> **Part 2**: We have provided a detailed comparison between PMP and ACM from the perspectives of motivation, methodology, theoretical insights.
>
> **Part 3**: We have corrected the repeated references.
>
> Warm regards,
>
> The Authors of Submission 9114.

---

> > ### Comment · Reviewer_8onS · 2023-11-23
> > **Thanks for the response**
> >
> > Thanks for the author's response especially the detailed discussion on the differences between ACM and the proposed PMP. These answers addressed my concerns, so I would like to increase my rating.

---

### Official Review · Reviewer_69ev · 2023-11-01

**Soundness:** 3 good
**Presentation:** 3 good
**Contribution:** 3 good
**Rating:** 6
**Confidence:** 3

**Summary:**

This paper describes a variant of GNN trained for graph fraud detection tasks.
Apparently, previously existing GNN's efficacy suffer due to label imbalance (a common occurrence with fraud data).
The authors' solution to this problem was in label-aware partitioning of aggregated contributions during message passing stage. Instead of aggregating contribution from all nodes from a given root node's neighborhood the proposed method would aggregate label-aware contributions separately which would include separate weight matrices for benign nodes and for fraud-related nodes.
The algorithm's theoretical examination shows that it independently learns an adaptive spectral filter for each node in the graph. The model also proved to outperform existing state-of-art solutions on publicly available datasets.

**Strengths:**

This paper is very well motivated and clearly written. The idea seems to be simple enough yet previous researchers' work focused on augmenting graph structure and label-augmented features and have not augmented message passing process only.

I found the visualization of the difference between the influence of fraud nodes in neighborhood of a given node and influence of benign nodes from the same neighborhoods to be very persuasive when using this metric for comparing minority class contribution with various GNN models.

**Weaknesses:**

I did not find any glaring weaknesses. I would just mention that it would be easier for a reader to follow if the same notation would not be used for different purposes.
h (page 15) - class homophili metric
h_{i}^{(l)} (page 3) - l-th layer hidden representation of v_i

**Questions:**

On page 4, equation 4: Shouldn't a small \alpha^{(l)}_{i} indicate that the model should treat unlabeled neighbors more similarly to benign nodes (instead of fraud nodes as written in the paper)?

On Figure 4 (page 8): Influence distribution: why GCN (and not R-GCN) was chosen as one of the methods for comparison? My understanding is that GCN cannot distinguish between diffenent relations and both Yelp and Amazon datasets include multiple relationship types. May be that is why GCN did better at T-Finance (with only one relation type)

---

> ### Author Response · Authors · 2023-11-18
> **Response to Reviewer 69ev**
>
> We appreciate your detailed suggestions for our work. We provide the response to your concerns point by point as follows:
>
> **Q1: It would be easier for a reader to follow if the same notation would not be used for different purposes.  $h$ (page 15) - class homophily metric $h_{i}^{(l)}$ (page 3) - $l$-th layer hidden representation of $v_i$.**
>
> **A1:** Thank you for pointing out the confusion caused by the overlapping use of the notation $h$. To address this, we have revised the notation to eliminate any ambiguity. The class homophily metric previously denoted by $h$ on page 15 will now be denoted by $\eta$ throughout the paper. We have marked these changes in blue in the revised manuscript.
>
> **Q2: On page 4, equation 4: Shouldn't a small $\alpha^{(l)}\_{i}$ indicate that the model should treat unlabeled neighbors more similarly to benign nodes (instead of fraud nodes as written in the paper)?**
>
> **A2:** Thank you for pointing out the inconsistency regarding the interpretation of the parameter $\alpha^{(l)}\_{i}$ in equation 4 on page 4. Upon re-examination, we agree that there was an error in our description of how $\alpha^{(l)}\_{i}$ influences the treatment of unlabeled neighbors in our model.
>
> As you correctly noted, a small $\alpha^{(l)}\_{i}$ should indeed indicate that the model treats unlabeled neighbors more similarly to benign nodes due to the larger weight placed on $\mathbf{W}\_{\text{be}}^{(l)}$. We have corrected the text in the manuscript to accurately reflect this relationship.
>
> **Q3: On Figure 4 (page 8): Influence distribution: why GCN (and not R-GCN) was chosen as one of the methods for comparison? My understanding is that GCN cannot distinguish between diffenent relations and both Yelp and Amazon datasets include multiple relationship types. May be that is why GCN did better at T-Finance (with only one relation type)**
>
> **A3:** Thanks for your deep insights. Our aim is to provide a comprehensive comparison across a spectrum of GNNs, ranging from the most basic GCN to the more advanced and fraud-specific model, BWGNN. These methods are selected as the most representative baselines for different classes of GNNs. We agree that the Yelp and Amazon datasets contain multiple relationship types and that GCN’s inability to differentiate between these may impact its performance. In the single-relational dataset T-Finance, the disadvantages of GCN are reduced because other methods no longer have the advantage of handling multiple relations, which is a logical result.
>
> We appreciate the suggestion that including R-GCN, which is designed to handle multiple types of relationships, could provide a more direct comparison for datasets with such complexity. To address this, we have conducted additional experiments with R-GCN, which we have included in the **Appendix E.3** of the revised manuscript. From the extended experiments with R-GCN in **Appendix E.3**, we include R-GCN due to its ability to handle multiple types of relationships within a graph. R-GCN extends the GCN model to account for different types of relations in the graph. Results illustrate the influence distributions for the multi-relational Yelp and Amazon datasets. It is evident that accounting for multiple relationships results in a rightward (positive) shift in the influence distributions.

---

> ### Author Response · Authors · 2023-11-22
> **Looking forward to your response**
>
> Dear Reviewer 69ev,
>
> Thank you once again for your insightful feedback on our submission. We would like to remind you that the discussion period is concluding. To facilitate your review, we have provided a concise summary below, outlining our responses to each of your concerns:
>
> **Part 1**: We have carefully checked the manuscript to ensure that all notations are now distinct and consistently applied throughout the paper.
>
> **Part 2**: We have corrected an inaccurate description of $\alpha$.
>
> **Part 3**: We compare our model with R-GCN and analyze their relationships.
>
> Warm regards,
>
> The Authors of Submission 9114.

---

### Author Response · Authors · 2023-11-23
**A Summary of Our Contributions and Responses**

Dear Reviewers and ACs,

As the discussion period draws to a close in the coming hours, we are grateful for the insightful comments and valuable suggestions provided by all reviewers. In the following, we would like to summarize the contributions and revisions of this paper again.

**Contributions:**

- **[High-level insight]** We advocate that the essence of effectively applying GNNs to graph fraud detection tasks lies in the ability to distinguish between neighbors during message passing. This perspective is ```interesting``` (_R 8onS_), ```well-motivated```, and ```simple enough yet previously unconsidered concept ``` (_R 69ev_). And also pave the way for future research in this area, as noted in our responses to Reviewers XqA8 and wN2Y.
- **[Simple yet effective model]** Our proposed PMP model innovatively partitions neighbors during aggregation using different weight matrices, a design that is ```well-designed```, ```straightforward```, ```easy to understand and implement```, ```effective``` (_R XqA8_), and also ```intriguing``` (_R wN2Y_). The introduction of a new visualization technique to showcase influence distribution has been commended as ```very persuasive``` (_R 69ev_) to validate the effectiveness of our model.

- **[Theoretical insight]** The model is supported by ```comprehensive analytical framework``` which is ```credible```(_R XqA8_).The theoretical insight from the perspective of spectral analysis is ```sound``` (_R 8onS_).

**Responses and Revisions**

- **[For reviewer 69ev]**
  * **Clarifications and corrections**: We have revised the paper to correct ambiguous notations, symbols, and descriptions for better clarity and accuracy.
  * **Extended experimental analysis**: Additional experiments have been conducted to explore and analyze the relationship between our model and R-GCN, providing a deeper understanding of their comparative performance.
- **[For reviewer 8onS]**
  * **Homophily and label imbalance**: We have enhanced the discussion to highlight the interplay between homophily and label imbalance in the context of fraud detection tasks.
  * **Comparative analysis with ACM**: A detailed analysis and comparison with similar work ACM are now included, offering insights from the perspectives of motivation, methodology, and theoretical underpinnings.
  * **Reference corrections**: We have addressed and corrected repeated references to ensure accuracy and completeness in our citation practices.
- **[For reviewer XqA8]**
  * **Paper structure and clarity**: Descriptions within the paper have been revised for clarity, and the overall structure of the sections has been reorganized for better coherence.
  * **Hyperparameter details**: Additional information regarding the detailed hyperparameter settings, particularly concerning the hidden layers of the neural networks, has been included.
  * **In-depth analysis on T-Social dataset**: A comprehensive analysis focusing on the homophily ratio, imbalance level, and common statistics has been conducted on the T-Social dataset. This analysis aims to provide a thorough understanding of our model's superior performance on this dataset.
- **[For reviewer wN2Y]**
  * **Time complexity and storage comparison**: We have provided a detailed comparison of the time complexity and storage costs between our model and other baseline models.
  * **Difference with BWGNN**: We highlight the distinct aspects of our model compared to BWGNN, especially from the perspectives of motivation and methodology.
  * **Label distribution analysis**: An analysis of the label distribution of labeled center nodes in benchmark datasets has been included. This analysis may addresses concerns regarding potential model failure and demonstrates the effective training of our model using the partitioning neighborhood approach.

All modifications have been highlighted in blue in our revised manuscript. Thanks again for your efforts in reviewing our work, and we hope our responses can address any concerns about this work.

Thanks

The Authors of Submission 9114.

---

### Meta-Review · Area_Chair_E6oy · 2023-12-12

**Metareview:**

The manuscript introduces an approach called partitioning message passing as a variant of GNN for graph fraud detection. The performance is demonstrated by comparing with previous approaches. The algorithm is also motivated from connections with spectral graph filtering. While some reviewer concerns are fully resolved during the discussion stage, after carefully examine the manuscript and discussion, the AE tends to recommend acceptance.

**Justification For Why Not Higher Score:**

Some reviewer concerns are fully resolved during the discussion stage, and the overall impact of the work seems limited as it applies only to a specific application area of GNN.

**Justification For Why Not Lower Score:**

The empirical performance of the introduced methodology looks convincing.

---

### Decision · Program_Chairs · 2024-01-16

Accept (poster)